# Speech and Language Therapy Service for Multilingual Children: Attitudes and Approaches across Four European Countries

**Theresa Bloder** [1,*]**, Maren Eikerling** [2,3]**, Tanja Rinker** [1] **and Maria Luisa Lorusso** [2]

1   Faculty of Languages and Literatures, Universitätsallee 1, Catholic University Eichstätt-Ingolstadt, 85072 Eichstätt, Germany; tanja.rinker@ku.de
2   Scientific Institute IRCCS E. Medea, 23842 Bosisio Parini, Italy; m.eikerling@campus.unimib.it (M.E.); marialuisa.lorusso@lanostrafamiglia.it (M.L.L.)
3   Department of Psychology, Piazza Ateneo Nuovo 1, University Milan-Bicocca, 20126 Milan, Italy
*   Correspondence: theresa.bloder@ku.de

**Abstract:** Educational equality and the reduction of discrimination are among the UN's Sustainability Goals. Previous studies as well as policy recommendations suggest that the extent to which these are implemented in the field of speech and language therapy for multilingual children depends on sufficient knowledge and material. To this end, an online survey was carried out with 300 Speech and Language Therapists (SLTs) from Austria, Germany, Italy, and Switzerland, investigating their attitudes and approaches regarding the service provision for multilingual children. Their responses were analyzed taking the SLTs' language background, experience, and country of origin into account. Results were interpreted in the context of country-specific SLT service-related policies and SLT training as well as migration history. There seems to be a gap between the SLTs' knowledge about the specific requirements for providing Speech Language Therapy (SLT) for multilingual children and their common practice, which—despite the continuous need of further training—points to sufficient awareness but a lack of materials or resources. We found experience in working with multilingual children to be the most influential factor on attitudes and approaches towards multilingualism. This suggests the importance of improving pre-exam and early-career professional experience to foster SLTs' development of mindful attitudes and appropriate approaches towards multilingualism in their clinical practice.

**Keywords:** multilingualism; developmental language disorder; speech and language therapy; sustainability; professional experience



## 1. Introduction

Children's academic and future professional achievements are known to depend, among other factors, on successful language acquisition [1,2]. Therefore, being able to detect and address atypical language development as early as possible has become increasingly crucial. Appropriate language competence is relevant in family and academic contexts in order to allow for the development of a cultural and social identity as well as cognitive development. This topic—among others—is addressed at an international level by the Sustainable Development Goals (SDGs) [3], 17 goals that were determined by the United Nations (UN) in 2015 aiming "to end poverty, protect the planet and ensure prosperity for everyone (...)". In this paper, special emphasis is placed on "3: Good health and well-being" and "4: Quality education" which are also related to "8: Decent work and economic growth" and "10: Reduce inequality".

Thus, this paper concerns the sustainability of approaches but also attitudes of speech and language therapy (SLT) professionals with respect to identifying developmental language disorders in children growing up multilingually. In order to gain an international

perspective, speech and language therapists (SLTs) from four different European countries were included.

## 1.1. Developmental Language Disorder (DLD)

Developmental Language Disorder (DLD) is a condition in which, according to the definition of the Diagnostic and Statistical Manual of Mental Disorders—DSM-5 [4], a child's language performance significantly differs from the performance of the majority of his/her peers without any underlying biomedical explanation. Different linguistic areas like phonology, syntax, or word finding and semantics can be affected by DLD to a varying degree [5]. As opposed to previous—more narrow—definitions of DLD as a disorder that is reflective of a child's intrinsic difficulty to acquire language (e.g., [6]), the CATALISE Consensus group [5] recently agreed that the presence of neurobiological and/or environmental risk factors does not rule out a diagnosis of DLD. The potential risk factors that are most frequently associated with DLD include, among others, a family history of language impairment, male gender, and a low level of parental education and/or socioeconomic status (SES; [7–9]).

## 1.2. Multilingualism

In general, multilingualism describes the circumstances in which a person is confronted with more than one language in his/her everyday environment [10]. Heritage language (also referred to as minority, family, home, or first language) is the language that a person uses or a child acquires at home, while the societal language (also referred to as majority, community, second language) is the language spoken by the majority of the society they live in. While the typical course of monolingual language development has been described in great detail, multilingual language development in comparison is far more heterogeneous as each multilingual child acquires his/her languages under diverse conditions, with highly individual language contact patterns, and, thus, with highly variable expressions of his/her language competences [11]. Abilities in two or more spoken languages can be at a similar level, i.e., balanced, but linguistic abilities in one or some languages may be better developed (language dominance) [12]. Many aspects of multilingualism influence language performance, such as (1) the onset, duration, quantity, and quality of exposure, (2) cross-linguistic influence specific to each language combination, and (3) dynamic and context-specific language use, which results in variance in language performance. Independent of the particular pattern of language exposure, children who acquire more than one language are not able to dedicate the same amount of time to each of them compared to monolinguals [13]; therefore, their language performance in either language alone cannot be expected to be comparable to that of their monolingual peers. In multilingual children it is thus crucial, but difficult, to distinguish poor language performance due to DLD from poor language performance that can be traced back to insufficient exposure [14].

## 1.3. Speech and Language Therapy in Multilingual Contexts

In many European countries, multilingualism is no longer an exception but the norm [15], which is also represented in the field of SLT: A.

Almost half of the children who receive SLT services grow up in multilingual settings (see [16]: 42.7%; [17]: 49% for Germany). Even though multilingual language acquisition differs from monolingual language acquisition [18], multilingualism per se is not to be regarded as a factor endangering language development [15,19]. In fact, the same proportion of diagnoses of DLD across all children (prevalence of 7% [20])—monolingual or multilingual—can be expected.

Nevertheless, the data of Lehti, Gyllenberg, Suominen, and Sourander [21] show that poor language development and academic performance are more frequently diagnosed among children of foreign-born parents. Similarly, De Lamo White and Jin [22] as well as Saenz and Huer [23] traced these differences between children born in monolingual

families and children born in families with an immigrant background back to inadequacies and biases in the assessment of linguistic abilities of multilingual children with diverse socio-cultural backgrounds. One possible explanation for the higher proportion of multilingual children supposedly showing SLT needs could be linked to the issue of low SES, presuming that lower SES is more prevalent among families with migration and, thus, a multilingual background. Indeed, SES may exert an influence on (multilingual) language development through two different mechanisms. The first mechanism is a general influence on language performance through varying degrees of richness and diversity in caregivers' speech [24–27], and the quality and amount of complex language input in the form of shared book-reading, story-telling, or varied social discourse relevant for the development of narrative competence [10]. Such input-related factors linked to low SES status coupled with multilingualism (the two factors act on language development independently from each other, as shown by Calvo and Bialystok [28]) can make it even more difficult to adequately identify language impairment, increasing the risk for misdiagnosis. The second type of mechanism is the effect of SES on the probability that a genetic predisposition manifests itself as a developmental disorder, through the action of epigenetic, nutritional, and other environmental variables [9]. Such effects may account for part of the increased prevalence of DLD in the multilingual population.

Yet, it can be said that despite the high proportion of multilingual children receiving SLT services, SLT practice is predominantly monoculturally and monolingually oriented [29]. This kind of practice can lead to either over- or underdiagnosis of DLD in multilingual children, overlooking clinically relevant language disorders or (which appears to be more common) providing SLT services to children with as yet insufficient skills in the societal language, which may lead to unnecessary stigmatization [30] as well as inappropriate allocation of health system resources [20].

Circling back to the topic of sustainability, these deficits in SLT services in particular relate to SDG "10: Reduced inequalities" [3], a topic that has not only been approached scientifically, but also politically through so-called policy papers, in which best-practice recommendations and guidelines for SLT services (i.e., diagnostics and intervention) for multilingual children are stated.

In accordance with the SDG 10 on the reduction of inequalities [3], the MULTI-SLI [31] position statement demands equal access to SLT services for mono- as well as multilingual children, but SLT service provision depends on the definition and identification of intervention needs. There is scientific consensus that DLD always affects all languages spoken by a child [5,14,31,32]. Therefore, oral skills need to be assessed in all the languages spoken in order to reliably distinguish DLD from variations in multilingual language acquisition due to unbalanced multilingual language input [14,22]. However, there is a disconnect between applied clinical practice and common opinion and theoretical knowledge when it comes to treating multilingual children. This is evidenced by the fact that children who live in homes where a heritage language is spoken are on average assessed and treated in the societal language only, even though SLTs declare not to agree with this practice [33]. Generally, most SLTs only deliver services in monolingual mode (i.e., in the societal language) [34,35]. Only very few SLTs provide multilingual assessment and intervention [36]. However, research has shown that children's improvements in the language targeted by the intervention (i.e., the societal language) often do not transfer to their heritage language [37,38]. This suggests that children's heritage language has to be targeted specifically in order for children to be able to communicate both within and outside the family context.

Several studies have focused on identifying the factors that hinder SLTs from applying what may be labeled the "best practice protocol" when working with multilingual children. Based on these studies we identified three main types of barriers to effective assessment and treatment of multilingual children with DLD.

### 1.3.1. Lack of Language Proficiency beyond the Societal Language

Several efforts have been made to address the issue of providing assessment and intervention in the societal language while at the same time accounting for children's L2-learner status. The BiSLI COST Action IS0804, for example, has developed so-called "language-universal" assessment tasks that are supposed to be independent of the language(s) a child speaks and Boerma and Blom [39] showed that performance on such language-universal tasks was indeed correlated with children's language proficiency. Another diagnostic approach is to rely exclusively on the societal language, but to refer to norms that were standardized specifically on the multilingual population (cf. German example: LiSeDaZ [40]). Nevertheless, Scharff Rethfeldt and colleagues [32] as well as Garraffa and colleagues [14] still highlight the need for better knowledge about the language-specific sequences in the acquisition of linguistic elements (e.g., phonemes) and structures (e.g., morphosyntax) and language-specific clinical markers for DLD in order to provide adequate assessment and intervention. Furthermore, the person administering a language test needs to have knowledge of intra-linguistic change (language attrition) and inter-individual (language change) processes [41]. For this reason, Chilla [42] recommends that the assessment of children's heritage language skills should be conducted by trained, native-language speaking professionals. However, this requirement poses challenges for clinical and educational staff due to a lack of linguistic knowledge of the various heritage languages potentially represented in all of their multilingual patients [14,15,32,43]. In fact, Roseberry-McKibbin and colleagues [44] found that the problem that occurred most frequently in service delivery to multilinguals was therapists' lack of knowledge of the child's heritage language. Only a small percentage of SLTs provide therapy in more than one language and even fewer are L1 speakers of two or more languages (7% in Friedrich and Knebel [34]; 8.3% in Lüke and Ritterfeld [16]; 9.4% in Williams and McLeod [35]). Additionally, the overlap between the languages spoken by the SLTs and the children on their caseloads is limited and therapists' knowledge of the languages primarily spoken by their patients is only available in isolated cases. Although one-third of the respondents in Wintruff, Orlando, and Gumpert [45] indicated that they had native-like language skills (in addition to German, i.e., the societal language) that could potentially be used diagnostically, in the majority of cases these were English (25%) and French (5%), languages that are represented in only 5% to 7% of their multilingual patients. Beyond the restrictions of lack of language proficiency in a language other than the societal language affecting assessment and treatment of DLD in multilinguals, many practitioners also feel limited in their abilities to provide information and instructions to children's caretakers [46] and many believe that, even though it should not be compulsory, it would be beneficial to SLTs to be fluent in more than one language [35]. Clinicians who speak several languages indeed (and rather obviously) reported assessing and treating children in all their languages more often than their monolingual colleagues [47].

### 1.3.2. Lack of Satisfactory Diagnostic Tools Available for the Multilingual Population

Many SLTs feel rather inadequately equipped with the diagnostic materials that are currently available, particularly for the multilingual population [16], and many try to cope by instead using informal procedures when assessing the speech and language development of children from multilingual backgrounds [35]. Some, for instance, use self-constructed or translated versions of instruments in the societal language to elicit utterances in the children's heritage languages, and only in isolated cases do SLTs use assessment tools that have been explicitly normed for monolingual children in the respective heritage language [45]. For the purpose of language assessment of their patients' heritage languages, many SLTs alternatively focus mainly on anamnestic data [34] or questionnaires, specifically designed for bilingual children's parents [48,49]. However, collecting this information is made difficult when the therapist and the caregiver(s) do not speak the same language. Difficulties in the delivery of this service were reported to especially arise when a child's primary caregiver (typically the person who is most knowledgeable of the child's (language)

development) was unable to speak the therapist's language, leaving the other parent to act as the family spokesperson and interpreter [46]. This communicative barrier is often maintained because the majority of SLTs do not have access to professional interpreters at all [47]. This is why position statements and policy papers recommend the use of indirect measures to gain information on the linguistic background of the child [14,31,32]. Many SLTs, in fact, refer to parental questionnaires in the family's native language (if available) [45]. Another clear recommendation is that language performance of multilingual children should not be compared to monolingual language learners [31]. At the same time, the number of standardized tests providing norms for multilinguals is insufficient and—even more serious—due to heterogeneity of the multilingual population [32], it is difficult to create these norm data [14]. When assessing children's language performance in the societal language, SLTs usually rely on diagnostic material designed and normed for monolingual children [34]. Unfortunately, SLTs normally (have to) rely on the assessment of only the societal language using tests that were standardized exclusively for monolingual children [16]. Still, since monolingual norms generally are unsuitable for multilingual children [15], these test results have to be interpreted with caution and should be limited to the assessment of the educational issues such as the child's school-readiness or his/her need for additional support at school, while they are of little use to address clinical, diagnostic issues. Lüke and colleagues [16] highlight that test scores obtained from material designed for monolinguals should only be considered as a reference point in the course of the diagnostic process and have to be related to different factors of multilingualism. Otherwise, SLTs risk misdiagnoses and erroneous decisions concerning a child's need for therapy [43]. On average, multilingual children receive speech therapy over a longer period of time than their monolingual peers and their rate of successfully completing SLT services is lower [50], which, in part, may be explained by SLTs' difficulties to quantifiably track their patients' intervention progress and outcome due to the paucity of available standardized and normed assessment material [36].

Even when the heritage language of the child to be assessed is not spoken by the SLT, computerized language screenings may facilitate the assessment of heritage language skills in multilingual children with the help of visual, audio, and video recordings [51]. The application of modern technologies in language diagnostics can have a motivating effect on the child to be tested due to the attractiveness of the testing material [52]. Possibly, computerized screening instruments that are administered online rather than in the presence of test administrators (i.e., remotely) may enable facilitated access to language assessment that does not require logistical effort [53]. However, by applying computerized assessments, there is less flexibility to address external circumstances, such as distractions, that might affect the child's performance [54].

Nevertheless, being able to assess the heritage language using computerized screenings is not sufficient. Assessing children's competences in the societal language is equally important and many SLTs and parents feel especially under pressure to treat children in the societal language in order to support children's school-readiness [36].

For both Italian- and German-speaking countries, computerized language assessment material has been developed to assess both a child's heritage and societal language competencies (e.g., for an assessment for Italian as the societal language "BaBIL—Prove per la valutazione delle competenze verbali e non-verbali in bambini bilingui: manuale e materiali" [51] and for German as the societal language "Evozierte Diagnostik grammatischer Fähigkeiten für mehrsprachige Kinder" [55]). While test items are presented automatically, SLTs have to evaluate children's responses in the heritage language based on the test manual. Not only nationally, but also internationally, there have been attempts to create material that is comparable across languages to disentangle DLD in multilingual children from typical multilingual language acquisition. One example is the native app "speakaboo" [56] that has been developed to assess articulatory and phonological skills in a wide range of heritage and societal languages.

SLTs are not only involved in the diagnosis and treatment of DLD, but also in advising caregivers (also referred to as caregiver counseling or consultation) and are thus responsible for providing recommendations about how parents can best support their children's language development. Garraffa et al. [14] highlight that language use within the family might change when being immersed into a different linguistic context but emphasize that heritage languages should not be abandoned. They also clearly state that multilingualism does not negatively impact language acquisition, even if a child has a language impairment. The majority of SLTs agree with this practice, as shown by Williams and McLeod [35] in their investigation of parent counseling. An encouraging finding was that the majority of their respondents indicated that they advised parents of multilingual children to speak the languages they know best to their children (61.6%). About a quarter said they usually give the advice to use the family's L1 only (25.6%), which, according to Jordaan [36], could be interpreted as a strategy to ensure the development of both languages (exposure to the societal language is ensured in any context outside of a family's home). Beyond that, only very few SLTs (7.0%) recommended the use of the societal language only or gave other advice (5.8%) [35].

### 1.3.3. SLTs' Preparedness and Confidence Assessing and Treating Multilingual Children

The gap between clinical practice and common opinion is also evident in SLTs' perception of their own preparedness and, thus, confidence to diagnose and treat multilingual children with DLD. Unfortunately, many SLTs believe that they do not have the necessary expertise to work with culturally and linguistically diverse families [57]. In a German study by Wintruff and colleagues [45] only 3% of SLTs felt that their training had adequately prepared them to work with multilingual children. Of the respondents, 80% said that the topic of multilingualism was covered only "very little" or "not at all" throughout the course of their training or studies [45]. In their 2005 study, Roseberry-McKibbin and colleagues [44] investigated how frequently different problems in SLT service delivery were encountered by clinicians in the USA. In terms of SLTs' knowledge about the phenomenon of multilingualism, they found that the lack of knowledge of developmental norms in children's primary language(s) and difficulties in distinguishing a language difference from an actual disorder were encountered most frequently [44]. These findings suggest that SLTs have a need for additional support and information about language-specific linguistic phenomena and their developmental trajectory in different languages as well as specific procedures for multilingual children in order to feel more confident in working with them. In a recent study in Germany, 43.9% of SLTs indicated feeling insecure about assessing multilingual children [34]. Half of the participants in this study stated that they had attended workshops on multilingualism, a finding that substantiates the connection that has been made between SLTs' engagement in the topic of multilingualism and their perception of their own competence when working with multilinguals [16]. Comparing the results from a 1990 survey with their more recent data that were collected 11 years later, Roseberry-McKibbin and colleagues found that in 1990, 23.6% of respondents had engaged in university courses addressing multilingualism, whereas in 2001, 73% of respondents indicated that they had been taking similar classes [44]. This may indicate an increase in the attention the topic of multilingualism has received in recent years but also the challenges that SLTs face when trying to adequately assess and treat multilingual children. Another factor that has been found to impact on SLTs' confidence in their cultural and language competence is a multilingual socialization (influenced by their country of origin's history of immigration, internal minority groups, and patterns of language use). Taken together, these findings suggest that SLTs are very much aware of the barriers and try to overcome them by acquiring more knowledge and expertise.

### 1.4. Country-Specific Differences

Although the policy papers mentioned above all make comparable recommendations on best-practice procedures in SLT services, country-specific (legal) conditions and

infrastructures may impede or at least influence their implementation across different countries. To this end, the following study was conducted to investigate SLTs' attitudes and approaches towards multilingualism in four different European countries (Austria, Italy, Germany, and Switzerland). The selection of these four countries had practical and theoretical reasons: (1) As part of an International Training Network (MultiMind), these four countries were easily accessible through network members and (2) the four countries represent two different languages, and, both between and within languages (three of the four countries are German-speaking), a variety of different contexts with respect to migration historical and current situation, organizational and educational regulations, and health service delivery policies. This allows us to disentangle the effects of the different socio-organizational variables from possible effects of linguistic differences. To be able to relate our findings to the respective therapeutic infrastructure, we will first provide information about country-specific conditions of SLT service.

### 1.4.1. Education and SLT Training

In Germany, several occupational groups are qualified to provide SLT services. The different groups hardly differ in terms of their rights and duties when carrying out their professional activities. They do differ, however, in the course of their education and training. Currently there are two courses of education available: (1) via a vocational school (three years of training) and (2) via university (3 to five 5 of training and studies; there are several Bachelor's and Master's programs available across Germany). Even if the latter has become increasingly popular in recent years, the majority of SLTs have received 3 years of training. However, due to the heterogeneity in available programs, it is difficult to determine how the various different professional qualifications are exactly distributed across German SLTs.

In Austria, all SLT training has been available only at the university level since 2010. Six universities of applied sciences currently offer SLP programs in Austria. Each of them only takes on a limited number of students each year (or every 2 years). Potential candidates have to pass an entrance exam before moving on in the selection process. Students who successfully defend their thesis and pass the final exam after 3 years of training are awarded the title Bachelor of Science. Therapists who finished their vocational training before 2010 and therapists who completed their studies at university belong to the same occupational group and are completely equal concerning the rights and duties of their professional practice. Currently there is only one institution that offers a Master's program in SLT in Austria that only a very limited number of SLTs complete.

In Switzerland, SLTs are exclusively trained in universities and universities of applied sciences within 3 to 5 years of studies. There are several Bachelor programs and one Master's program. However, the large majority of SLTs in Switzerland has not obtained a Master's degree.

In Italy, SLTs are trained in a 3-year university program in SLT and can continue for 3 additional years to reach a Master's degree and extend their curricula in terms of health management, teaching, and research skills. In order to get accepted in an SLT university program, applicants have to pass an entrance exam. All the main universities offer both 3-year and master-level courses, although the large majority of SLTs in Italy choose the former option.

### 1.4.2. Referral Policy

If a need for SLT treatment is identified, in Austria, Germany, and Italy a prescription or referral (terminology depending on the country) from a doctor (e.g., ear-nose-throat (ENT), pediatrician, child neuropsychiatrist, or orthodontist) initiates therapy. Generally, a referral is made for a set number of therapy sessions (typically 10 sessions). After that, if necessary, a new specialist's referral is required.

In Switzerland, children's need for therapy is commonly determined by SLTs through assessment centers in schools. No medical referral is required. Therapy is usually initiated and provided within the school context. The duration of the SLT intervention is based

on each child's individual needs (there is no predetermined number of therapy sessions). A re-evaluation of the need for therapy takes place after about half a year of SLT service provision.

1.4.3. Therapy Costs and Insurance Coverage

In all four countries health insurance is obligatory. However, whether and to what extent the costs for SLT services are covered differs among countries.

In Germany, for patients who are younger than 18 years of age statutory medical insurance companies provide full cost coverage without a deductible. For people with private insurance (i.e., public servants and self-employed persons, people with medium to high SES), a small deductible might remain. Generally, families are not financially burdened when making use of SLT services.

In Austria, there are only a few SLTs whose service is covered entirely by the insurance companies. However, for the majority of SLTs in Austria patients sustain the entire service costs but may apply for a partial reimbursement from their health insurance company. Additionally, in order for insurance companies to (partially) cover the costs for the SLT services, the approval of a senior medical authority from the respective health insurance is necessary, based on a specialist's referral and an SLT's diagnostic report. Ultimately, a financial burden of between 30 to 60 euros per therapy session remains at the families' expenses.

In Switzerland, unlike in Germany and Austria, SLT services for children with speech and/or language disorders are not subject to the healthcare system but to the educational system. The majority of SLTs are directly employed by schools and receive a set monthly salary. In addition, SLTs may also work in free practices. In these cases, the therapy costs are settled between the therapist and a reimbursement office. Families do not have to pay for SLT services. Moreover, families are also entitled to make use of SLT services even if there is no medical indication (cf. as a measure of language support). In this case, however, the therapy costs must be borne entirely by the families.

In Italy, treatment in private centers accredited with the National Health Service (NHS, in Italian: Servizio Sanitario Nazionale (SSN)) or in public services is free of charge (except for a small contribution called "ticket", the amount of which varies between regions) for the patient and his/her family since the costs are covered by the national NHS. Costs for treatment in private clinics or practices instead are not covered by the NHS. Many families choose private treatment in spite of higher costs to shorten waiting times at NHS (or accredited) services.

1.4.4. Immigration Census Data and History

Data provided by the statistical office of the European Union [58] show that, in general, all four countries have relatively high proportions of foreign-born residents. The highest proportion of foreign citizens can be observed in Switzerland, where non-nationals account for almost 30% of the total population. In the other three countries, foreign citizens make up between 10–20% of the resident population (see Table 1), with the lowest proportion of immigrants found in Italy.

**Table 1.** Foreign-born population by country of birth as of 1 January 2020. Numbers include persons born in another EU Member State and persons born in non-EU Member States [58].

| Country | Foreign-Born Population [1] | |
| --- | --- | --- |
| | **Thousand** | **% of the Population** |
| Germany | 15,040.7 | 18.1 |
| Italy | 6161.4 | 10.3 |
| Austria | 1760.6 | 19.8 |
| Switzerland | 509.7 | 29.2 |

[1] Note that the numbers reported do not contain all people of foreign descent but with citizenship of one of the respective countries. Actual numbers of people with migration background are thus expected to be higher.

The sociological and historical characteristics of migration phenomena are very different among the four countries.

In the late 19th century, industrialization and the banking industry led to a boom in Switzerland's economy that was attractive to many migrant workers. When free movement of the population was established with Switzerland's neighboring countries the number of foreign residents doubled but fell again during World Wars I and II. However, since the end of World War II, immigration to Switzerland again has been on the rise, mostly driven by the favorable economic climate of the country. While Switzerland has endorsed the agreement on the free movement of people, one of several bilateral agreements with the European Union, immigration from non-EU countries is limited to skilled workers who are urgently required and are likely to integrate successfully in the long term [59]. Among the main countries of citizenship of Switzerland's foreign population are Italy, Germany, Portugal, and France ([58]; see Table 2).

Over the past decades, Germany has been subjected to several distinct waves of immigration. These include, among others, the resettlement of ethnic Germans from Eastern Europe after World War II and Germany's guest worker program of the 1950s to 1970s. Germany also took in a large number of refugees from the Yugoslav wars in the 1990s and most recently experienced the country's sharpest spike of immigration by asylum seekers from Syria, Afghanistan, and Iraq fleeing from war scenarios in their home countries. Based on data provided by Eurostat [58], the ethnical groups currently most prevalent among German immigrants are from Turkey followed by Poland, Syria, and Romania (see Table 2).

In the early 1960s, about only 1.4% of Austria's population was of foreign origin. That number soon increased due to targeted recruitment of workers from the former Yugoslavia and Turkey in the 1970s. After a few years of stagnating immigrant numbers, there was another wave of immigration in the early 1990s when Austria, like Germany, took in refugees from the former Yugoslavia (especially from Bosnia and Herzegovina, Serbia, and Kosovo). With the turn of the millennium, Austria experienced further growth of its migrant population, mainly due to increased immigration from the countries of the (enlarged) European Union. Most recently, in the years from 2014 to 2017, the admission of a considerable number of refugees and asylum seekers from Syria (among other countries) led to another growth of the numbers in the migrant population [60]. As of 2020, Austrian immigrants' most prevalent countries of origin are Germany, Romania, Serbia, and Turkey ([58]; see Table 2).

As opposed to the other three countries, Italy's history of immigration is a more recent phenomenon. In fact, Italy, for most of its history from unification onwards, has been a country of emigration and it is estimated that up to the late 1970s more than 24 million people left the country. In the early 1990s, Italy for the first time had to deal with a big wave of immigrants coming from Albania (caused by the collapse of the Eastern bloc) but immigration only became a defining phenomenon of Italian demography in the early 2000s with the onset of the Arab Springs and the main countries of embarkation of migrants coming from North Africa. A large community of Chinese immigrants lives in specific regions of Italy, and several people have come from Eastern countries such as Ukraine and Poland or Russia in the last decades to work in Italian families as helpers for the elderly. Since 2014, with the outbreak of the second civil war in Libya and the Syrian refugee crisis, Italy has experienced a new spike in the arrival of immigrants [61]. Today, among the most prevalent countries of origin for Italian immigrants are Romania, Albania, and Morocco ([58]; see Table 2).

**Table 2.** Main countries of citizenship of the foreign population as of 1 January 2020. Numbers are displayed as a percentage of the total foreign population [57].

| Germany | | Italy | | Austria | | Switzerland | |
|---|---|---|---|---|---|---|---|
| **Country of Citizenship** | **%** | **Country of Citizenship** | **%** | **Country of Citizenship** | **%** | **Country of Citizenship** | **%** |
| Turkey | 12.7 | Romania | 22.7 | Germany | 13.6 | Italy | 18.0 |
| Poland | 7.4 | Albania | 8.4 | Romania | 8.4 | Germany | 7.8 |
| Syria | 7.3 | Morocco | 8.2 | Serbia | 8.3 | Portugal | 7.3 |
| Romania | 6.8 | China | 5.7 | Turkey | 8.0 | France | 5.3 |
| Italy | 5.7 | Ukraine | 4.5 | Bosnia Herzegovina | 6.6 | Spain | 4.2 |
| Other | 60.1 | Other | 50.4 | Other | 55.2 | Other | 57.5 |

*1.5. Research Goals and Hypotheses*

The aims of the present study were:

- to explore SLTs' beliefs and approaches towards treating multilingual children both within and comparing the four countries through an online questionnaire;
- to assess how variables that were explicitly addressed in the questionnaire and that could differ both within and across countries affect attitudes and approaches in SLT services for multilingual children, precisely:
  - SLTs' overall professional experience (measured in years),
  - Experience in specifically working with multilingual children,
  - Educational background (BA vs. MA), indicated by respondents' years of education,
    (This variable was taken into consideration for the Italian sample only since the educational/training system for SLT in Italy is more homogeneous on a national level (3 years of BA compulsory; 2 additional years of MA optional).)
  - Respondents' personal language background (being monolingual vs. multilingual),
    (This variable was only taken into account for the subgroup of German-speaking SLTs (working in Germany, Austria, or Switzerland) to investigate how SLTs' beliefs regarding multilingualism influence their approaches in terms of applied procedures in the clinical practice. Since these countries compared to Italy have a larger proportion of multilingual citizens, due to both historical (migration) and educational factors (only about 14% of the overall population speaks English versus 40% in Austria and 31% in Germany, with only four additional language groups exceeding 1%, not including any language from migrant contexts, versus eight additional languages in Austria and seven in Germany, including various minority languages from migration contexts) [62]).

More specifically, and based on the sources reviewed above, or—in the absence of previous literature—on what we felt were the logical consequences of the latter, or the optimal way to face the challenges described, we expected that:

1. The large majority of SLTs acknowledge the importance of taking into account aspects of the child's heritage language and culture when treating multilingual children and providing parent training.
2. The large majority of SLTs is aware of, and—presuming that existing tools are reasonably applicable and easily accessible—optimally makes use of, a series of tools that facilitate SLT service provision for multilingual children (i.e., multilingual and computerized solutions).
3. Differences among the countries reflect the organizational, sociological, and historical differences described above but are mediated by SLTs' experience level in working with multilingual children.

4.  SLTs' experience in working with multilingual children positively correlates with greater awareness of the need for multilingual approaches and with better knowledge of available tools specifically designed for multilingual children.
5.  A longer duration of overall SLT practice (professional experience in years) is linked with more multilingually oriented approaches in SLT service provision.
6.  A longer duration of SLT education/training is linked with more multilingually oriented approaches in SLT service provision.
7.  Multilingual SLTs attribute more importance to multilingual approaches in SLT service for multilingual children, due to their greater awareness (and personal experience) of cross-linguistic influence and facilitation processes occurring across languages.
8.  Knowledge of special diagnostic material is associated with its actual use while providing SLT services for multilingual children.
9.  SLTs' who declare that DLD is not likely to be exacerbated by multilingualism (i.e., second language acquisition or cross-linguistic influence) are more open to a linguistically diverse language environment.
10. SLTs who favor the use of heritage language at home would be more likely to adopt multilingual approaches and are more open to multilingual material (including information and communications technology (ICT) solutions) in SLT services.
11. SLTs who have more experience and/or more extensive education and/or attribute more relevance to the multilingual background of their patients would also be more aware of the potential impact that multilingualism may have on both therapy and parent consultation.

## 2. Material and Methods

### 2.1. Participants

The target group of this study was defined as certified Speech and Language Therapists and other occupational groups working in the field of Speech and Language Therapy (e.g., speech therapists, clinical linguists, or state-certified breath, speech, and voice teachers), who diagnose and conduct therapy with children with developmental language disorder. Participants were either based in Italy ($n = 103$) or in one of three German-speaking countries, namely, in Germany ($n = 85$), Austria ($n = 45$), or Switzerland ($n = 67$). Altogether then, 197 respondents worked in German-speaking countries.

### 2.2. Questionnaire Design

The data were collected by means of a questionnaire, which was originally created and piloted in Italy and consisted of 24 questions. This Italian questionnaire was translated and adapted to German. A back-translation procedure was applied after the construction of the questionnaires, highlighting slight differences between the two versions of five of the questions (Q4, Q11, Q12, Q22, Q23). Such differences were discussed but it was not possible to find a German perfect equivalent to the Italian questions. In order to collect information about locally relevant issues, one question was added to the Italian version (concerning training pathways) and four questions were added to the German version (concerning years of professional experience and multilingual background of the SLT and her/his caseload composition) of the questionnaire, which led to a total number of 25 questions in the Italian and 28 questions in the German version. Overall, the questions addressed participants' general beliefs towards multilingualism in SLT, their experience, and their usual practice.

The questionnaire comprised 24 multiple-choice questions, two open questions, and three closed (yes–no) questions. In terms of its content, the questionnaire more specifically covered four thematic areas: (1) participants' prior experience with child multilingualism, (2) participants' beliefs towards approaches to DLD in multilingual children, (3) participants' personal clinical practice and their individual approaches to procedures for multilingual children with DLD, (4) barriers to effective assessment or treatment of multilingual

children with DLD. A full list of the questions (Q1–Q29) and response options in their English version is reported in the Appendix A (Table A1).

### 2.3. Data Analysis

Data were analyzed by means of IBM SPSS Statistics v.20. The prevalence of certain responses within the overall response distribution was assessed through chi-square goodness-of-fit analyses. In a second step, associations between the variables were assessed by chi square tests of independence. Since some of the variables provided answers that could be led back to an ordinal scale (all the questions for which possible replies were "never, rarely, sometimes, yes" or the similar, with three or four different ordinal levels), the reciprocal effects of such variables were further characterized through ordinal-x-ordinal logistic regression statistics such as Somer's D and gamma. First of all, the results were analyzed in the whole sample in order to describe the general attitudes and practices of SLTs in the group. Then, specific analyses were conducted to check the effects of the variables that were hypothesized to act as independent variables on other responses, following the hypotheses described above. Whenever a significant effect of a variable emerged, further analyses were performed to describe responses in the different subgroups. A more exploratory approach was followed for country-related differences, for which no clear, a priori hypotheses had been formulated but for which post hoc interpretations were proposed in the discussion, taking into account the differences in the countries as described in the introduction. For the five questions for which differences between Italian and German did not allow for a literal translation, preliminary analyses were performed to identify a possible, general effect of linguistic formulation. Whenever a significant effect of the linguistic version emerged, the results were analyzed separately for Italian-speaking SLTs and SLTs from the three German-speaking countries.

### 2.4. Procedure

In Italy, the questionnaire was initially distributed as a paper-and-pen version in July to September 2019 among the Speech and Language Therapists at the scientific institute IRCCS E. Medea, Bosisio Parini. Thirteen anonymous responses were collected and analyzed. The questionnaire was then implemented online using Google Forms and distributed between March and April 2021 with the help of FLI (Italian Federation of Logopedists) via social media. Invitations were also sent to, and links were shared within, groups of known members of SLT groups, even if responses were anonymously collected. This procedure was chosen to maximize privacy protection but also discourage misuse/improper use of the links by non-SLTs. Overall, 103 Italian SLTs participated in the study. All completed forms were included in the study (response to each of the questions was required to proceed through the online form).

Across the three German-speaking countries, the questionnaire was advertised online via the respective national professional associations (dbl and dbs in Germany, logopädieaustria in Austria, DLV in Switzerland), other mailing lists (e.g., SES interdisziplinär, alumni list Klinische Linguistik Bielefeld University), social media, and word of mouth between October 2020 and January 2021. In sum, 265 responses were collected. A total number of 197 responses were included in the analysis after the exclusion of (1) incomplete responses, $n = 54$ and (2) responses from participants with no prior professional experience (i.e., students), $n = 14$. Overall, among the German-speaking SLTs, 43.2% ($n = 85$) were from Germany, 22.8% ($n = 45$) from Austria, and 34.0% ($n = 67$) were from Switzerland.

### 2.5. Declaration of Consent and Data Storage

Data were collected in a completely anonymous form. Before starting the survey, all respondents confirmed that they were willing to participate in the study and agreed to the storage and processing of their anonymous responses as well as to their use for scientific purposes and possible publications.

## 3. Results

### 3.1. SLTs' Characteristics

Among all SLT respondents ($N = 300$), only 7% declared that they had no experience at all in working with multilingual children (Q1; $X^2$ (3, $N = 300$) = 34.85, $p < 0.001$), while 12.7% declared that they had some experience, and 80.3% regularly provided services for multilingual children with DLD.

Among SLTs from German-speaking countries ($n = 197$; the question about years of professional experience was only included in the German questionnaire), 20.8% had less than 5 years of overall professional experience in providing SLT services, 25.9% had between 5 and 10 years, 21.8% between 10 and 20 years, 31.5% had more than 20 years of overall professional experience in providing SLT services to children (Q25). Furthermore, of all German-speaking SLT respondents, only 17.8% indicated that they could speak a language other than German at native-like level (Q27). There were 10.2% who, in addition to German, spoke languages that are typically taught in school settings (i.e., English and French) or languages that are present in neighboring countries and, thus, border regions (e.g., Dutch), while 7.6% spoke a language that could be considered more prevalent among the migrant community (e.g., Turkish, Farsi, Polish) (Q28). The distribution of the rate of multilingual children with DLD with respect to SLTs' total caseload (Q26) is displayed in Table 3. Information on caseload was only investigated in the German version of the questionnaire (Q26) and provided by $n = 154$ German-speaking respondents. These answers belong to a separate database and have been analyzed only with respect to their relationship with experience (Q1) and country, while further, more in-depth analyses will be performed on the German-speaking respondents only.

**Table 3.** Distribution of multilingual children with DLD among German-speaking respondents.

| | What Percentage of Children you Diagnose/Treat for DLD are Multilingual? | | | | | |
|---|---|---|---|---|---|---|
| | *0–5%* | *6–25%* | *26–50%* | *51–75%* | *76–95%* | *96–100%* |
| proportion of German-speaking SLT respondents ($n = 154$) | 5.8% | 22.1% | 24.7% | 26.6% | 16.2% | 4.6% |

As to the question concerning SLT training (Q29; only included in the Italian questionnaire), among the Italian SLTs ($n = 103$), 82.5% had completed a Bachelor's degree, 9.7% had completed a Master's program, and 7.8% had a different educational background.

Independently of country of workplace, experience in working with multilingual children, duration of SLT training, and SLTs' own language backgrounds, the results concerning SLTs' attitudes and approaches to multilingualism and DLD are described below.

### 3.2. SLTs' Attitudes in the Whole Sample

The significant majority of SLTs are supportive of heritage language use (Q5; $X^2$ (3, $N = 300$) = 188.81, $p < 0.001$), while 89.7% believe that children's communication partners should always speak in the language they know best, and only 10.3% think that for multilingual children with DLD language input should be reduced to a single language (both inside and outside the family environment). Furthermore, the majority of all SLT respondents think that different approaches are needed in the diagnosis and treatment of multilingual children compared to monolingual children (Q2; $X^2$ (3, $N = 300$) = 215.12, $p < 0.001$); more precisely, SLTs think that different approaches are always (70.0%)/sometimes (28.7%)/never (1.3%) needed for multilingual children. Furthermore, the majority of SLTs think (34.7%) and tend to think (24.3%) that DLD is independent of multilingual language acquisition (Q4; $X^2$ (3, $N = 300$) = 34.85, $p < 0.001$). Additionally, the majority of all SLT respondents also indicated that intervention should take both a child's heritage and societal language into account (Q3; $X^2$ (3, $N = 300$) = 85.92, $p < 0.001$) (see Table 4).

**Table 4.** Distribution of SLTs' responses to Q3 and Q4, concerning general attitudes towards multilingualism.

| | A | More A Than B | More B Than A | B |
|---|---|---|---|---|
| Q3 Do you think that (A) the therapy of DLD in multilingual children should be limited exclusively to the societal language, or that (B) the child's heritage language should also be taken into account? | 3.0% | 25.4% | 36.8% | 34.8% |
| Q4 Do you believe that (A) DLD is independent of speaking a second language, or that (B) multilingualism may impact the manifestation of DLD? | 34.7% | 24.3% | 29.3% | 11.7% |

As shown in Table 5 a significant majority (63.3%, $X^2$ (3, $N = 300$) = 294.26, $p < 0.001$) of the SLT respondents think that it is useful to compare the child's language performance in the different languages spoken (Q6). More specifically, most participants declare that it is useful to compare children's performance in the various linguistic domains (Q11, phonology: 79.7%; Q12, morphosyntax: 77.7%; Q13, vocabulary: 76.3%; see Table 5 for complete results) in their heritage language with their performance in the societal language ($X_s^2$ (3, $N = 300$) = 431.60–490.43, all $p_s < 0.001$). Since for two of these questions the German and the Italian translation of the question differed slightly, the responses were preliminarily compared between the two versions. While for Q11 no significant difference emerged ($p > 0.05$), we found a small but significant difference for Q12 ($X^2$ (3, $N = 300$) = 7.90, $p = 0.048$; Cramer's *V/Phi* = 0.16). While 1.9% of the German-speaking SLTs declared that they did not find it useful to check whether the syntactic and morphological structures that cause difficulties to the child in the societal language are affected in the heritage language, 5.6% of the Italian SLTs declared that they did not find it useful to check whether the syntactic and morphological structures that cause difficulties to the child in the societal language exist in the heritage language (see Table 6 for German-speaking and Italian SLTs' responses reported separately).

**Table 5.** Distribution of SLTs' responses concerning the usefulness of comparison of different aspects of children's heritage and societal language performance.

| Do You Think That It Would Be Useful/Helpful to ... | No | Rarely | Sometimes | Yes |
|---|---|---|---|---|
| Q6 ... compare a child's language performance in his/her heritage and societal language? | 2.0% | 4.7% | 29.1% | 63.4% |
| Q11 ... check whether the phonemes that cause difficulties to the child in the societal language are present/are also affected in the heritage language? | 2.0% | 3.3% | 15.0% | 79.7% |
| Q15 ... have a summary chart of the phoneme inventory of the child's heritage language? | 1.7% | 3.7% | 17.7% | 77.7% |
| Q12 ... check whether the syntactic and morphological structures that cause difficulties to the child in the societal language are present/are also affected in the heritage language? | 4.3% | 1.7% | 17.0% | 77.7% |
| Q16 ... have a summary table of the main syntactic structures and constructions in the child's heritage language? | 2.0% | 1.7% | 14.3% | 82.0% |
| Q13 ... check whether words that the child uses semantically/lexically incorrectly in the societal language are similar or very different from the heritage language? | 4.0% | 4.3% | 15.3% | 76.3% |

**Table 5.** *Cont.*

| Do You Think That It Would Be Useful/Helpful to … | No | Rarely | Sometimes | Yes |
|---|---|---|---|---|
| Q17 … have an overview table with a list of the most important prepositions (translations & usage) in the child's heritage language? | 4.0% | 6.3% | 18.3% | 71.3% |
| Q20 … give the child and his/her family some homework to practice in their heritage language? | 6.7% | 7.7% | 41.7% | 43.7% |
| Q21 If such tasks were available to you, would you use them? | 4.7% | 5.7% | 26.7% | 63.0% |
| Q18 … assess the level of proficiency in the child's heritage language with computerized tasks? | 5.3% | 5.3% | 32.7% | 56.7% |
| Q19 If such tasks existed, would you use them when working with multilingual children? | 5.0% | 4.7% | 36.0% | 54.3% |

**Table 6.** Distribution of SLTs' responses to Q12 in the two language versions.

| | No | Rarely | Sometimes | Yes |
|---|---|---|---|---|
| Q12 (German version) Do you think it would be useful to check whether the syntactic and morphological structures that cause difficulties to the child in the societal language are also affected in the heritage language? | 5.6% | 2.5% | 19.3% | 72.6% |
| Q12 (Italian version) Do you think it would be useful to check whether the syntactic and morphological structures that cause difficulties to the child in the societal language are present in the heritage language? | 1.9% | 0% | 12.6% | 85.4% |

### 3.3. SLTs' Clinical Approaches in the Whole Sample

The significant majority of our SLT respondents draw (32.2%) or tend to draw (35.5%) their comparisons of children's language performance between their two languages based on information provided by the parents, whereas only 17.6% are usually able to/14.5% are more often able to observe children's language behavior in both languages directly (Q7; $X^2$ (3, $N$ = 290) = 38.41, $p$ < 0.001).

While only about one-third (35.7%) of SLT respondents was not, two-thirds (64.3%) were aware of diagnostic material specifically designed for multilingual children (Q8; $X^2$ (3, $N$ = 300) = 24.65, $p$ < 0.001). However, only a few of our SLT respondents (18.1%) regularly use such material when assessing multilingual children's language status (Q9: $X^2$ (3, $N$ = 300) = 78.89, $p$ < 0.001), while 22.4% declare to use such material sometimes and 13.0% rarely; the significant majority of SLT respondents (46.5%) declared to not use them at all.

All SLT respondents further declared they considered it useful to consult material for different heritage languages to support their work with multilingual children, as shown in Table 5 (Q15, summaries of phonemic inventories; Q16, overview of main syntactic structures and constructions; Q17, overview table with a list of the most important prepositions; $X^2$ (3, $N$ = 300) = 357.68–532.35, $p$ < 0.001). Additionally, SLTs think it would be useful to have (Q20; $X^2$ (3, $N$ = 300) = 152.04, $p$ < 0.001) and declare they would use (Q21; $X^2$ (3, $N$ = 300) = 268.08, $p$ < 0.001) ready-made material in the heritage language for at home exercise. The majority of participants also declared that it would be useful to have computerized tasks to assess the level of proficiency in the child's heritage language (56.7% "yes", 32.7% "sometimes"), $X^2$ (3, $N$ = 300) = 220.21, $p$ < 0.001. Most SLTs further declared that they would in fact use these tasks (54.3% "yes", 36.0% "sometimes"), $X^2$ (3, $N$ = 300) = 215.39, $p$ < 0.001.). SLT respondents think that differences in the languages spoken by the child and the therapist may always (31.0%)/sometimes (48.0%)/rarely (10.7%)/never (10.3%) influence the quality of the intervention (Q22; $X^2$

(3, $N = 300$) = 118.27, $p < 0.001$). Since for this question two slightly different wordings were provided for the German-speaking participants compared to the Italian participants, we performed a chi-square test of independence to compare SLTs' responses in the two versions. The comparison between the German and the Italian version yielded a significant result ($X^2$ (3, $N = 300$) = 26.36, $p < 0.001$, Cramer's $V/Phi$ = 0.30). We therefore report SLTs' responses for the two language versions separately: 23.9% of the German-speaking SLT respondents believed that the fact that the therapist and the child speak different languages always influences the quality of the intervention as opposed to the rest who believed that it did only sometimes (47.7%)/rarely (15.7%)/never (12.7%). Within the group of Italian respondents, 44.7% believed that linguistic differences always influence the quality of the intervention, compared to the remaining Italian SLTs who believed they did only sometimes (48.5%)/rarely (0.97%)/never (5.8%). While only a few SLT respondents believed that differences in the languages spoken by the child and the therapist did not influence the quality of parent consultation (5.7% "never", 3.3% "rarely"), almost the entire group of respondents (48.3% "always", 42.7% "sometimes") believed that they did (Q23; $X^2$ (3, $N = 300$) = 203.97, $p < 0.001$). However, similar to Q22, the two slightly different versions for Italian and German were found to produce significantly different results in SLTs' responses ($X^2$ (3, $N = 300$) = 10.29, $p = 0.016$, Cramer's $V/Phi$ = 0.19) and were analyzed separately. In general, 45.2% of the German-speaking SLT respondents believed that the fact that the therapist and the child speak different languages always influences the quality of parent consultation as opposed to the rest who believed that it did only sometimes (42.1%)/rarely (5.1%)/never (7.6%). Within the group of Italian respondents, 54.4% believed that linguistic differences always influence the quality of parent consultation, compared to the remaining Italian SLTs who believed they did only sometimes (43.7%)/rarely (0.00%)/never (1.9%).

Concerning SLTs' beliefs about whether or not cultural differences between the therapist and the child's family may impact the quality of parent consultation (Q24), SLTs declared that such differences would never (7.3%)/rarely (8.7%)/sometimes (45.5%)/always (38.5%) have an effect on their service of parent consultation (Q24; $X^2$ (3, $N = 300$) = 140.88, $p < 0.001$).

In order to determine whether or not there were significant associations between variables related to experience and any of the SLTs' responses, we applied chi square tests of independence (see Sections 3.1–3.4).

Experience has a significant impact on the SLTs' beliefs about the influence of differences in the languages spoken by the child and the therapist on the quality of intervention (Q22): The more experienced the SLTs, the less convinced they are that multilingualism may influence the quality of the intervention ($X^2$ (6, $N = 300$) = 21.18, $p = 0.002$, Cramer's $V = 0.188$, $Phi = 0.226$, Somer's $D = -0.327$, $p < 0.001$). While all SLTs who have never provided therapy for multilingual children expect an impact on the quality of their service, most experienced SLTs are less polarized in their opinions regarding the influence of language differences on therapy provision (see Figure 1).

However, across all experience levels, the majority of SLTs believe that to a certain extent language differences between the child and the therapist play a role in the quality of intervention. Considering the topic of parent consultation, we did not find any effect of experience on SLTs' perception of the influence of speaking different languages (Q23; $p = 0.428$) or of cultural differences (Q24; $p = 0.082$).

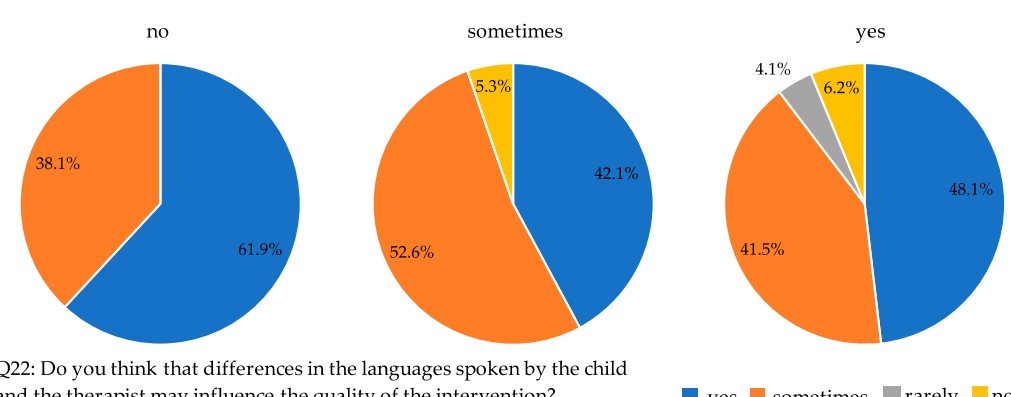

Q1: Have you ever provided therapy to multilingual children with DLD?

Q22: Do you think that differences in the languages spoken by the child and the therapist may influence the quality of the intervention?

**Figure 1.** Distribution of SLTs' responses concerning quality of SLT services according to the SLTs' level of experience.

### 3.4. Country-Related Effects

Of the respondents of our sample, 103 lived in Italy (34.3%), 85 (28.3%) in Germany, 45 in Austria (15.0%), and 67 lived in Switzerland (22.3%). We found a highly significant association between the amount of experience in working with multilingual children and country, $X^2$ (6, $N = 300$) = 79.56, $p < 0.001$. This was a medium to large effect (Cramer's $V = 0.36$, $Phi = 0.52$). Of all four countries, Italian SLTs seem to be the least experienced in working with multilingual children. While in Austria, Germany, and Switzerland, more than 90.0% of respondents indicated to have experience in working with multilingual children, this applies to only 52.4% of Italian respondents. Furthermore, 15.5% of Italian SLTs stated that they had no such experience at all (see Figure 2).

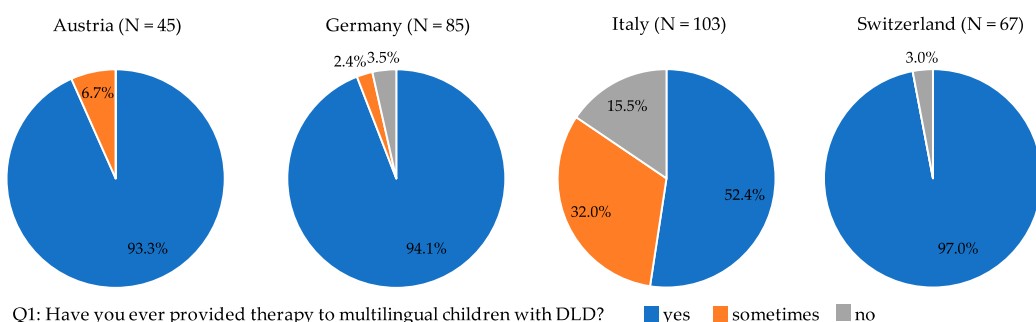

Q1: Have you ever provided therapy to multilingual children with DLD?

**Figure 2.** Distribution of SLTs' experience in working with multilingual children with DLD across the four countries.

In the subgroup of 154 of the SLTs from the three German-speaking countries for whom a question on caseloads (Q26) was included in the questionnaire, we found a non-surprising, significant relationship between SLTs' general experience in working with multilingual children (Q1) and the proportion of multilingual children on their caseloads (Somer's $D = 0.510$, $p = 0.03$). The more experience SLT respondents had, the greater the number of multilingual children on their caseloads. The three German-speaking countries differed from each other in multilingual caseloads (Q26, not included in the Italian questionnaire), as suggested by a significant effect of country within German-speaking SLTs' ($n = 154$) on this variable ($X^2$ (10, $n = 154$) = 20.00, $p = 0.029$). Austrian SLTs seem to have fewer multilingual children on their everyday caseloads compared to SLTs practicing in Germany and Switzerland. In Austria, the proportion of SLTs who have hardly any multilingual children on their caseloads (ranging from 0 to 5%) is larger (14.7%) compared to Germany (1.5%) and Switzerland (5.6%). This trend in low representation of multilingual children in SLT service provision in Austria is also reflected when comparing the proportion of highly specialized multilingual caseloads (ranging from 96% to 100%)

among those countries. While in Germany and Switzerland some (but very few) SLT respondents are highly specialized on multilingual caseloads (Germany 4.6%, Switzerland 7.3%), no such specialization exists for any of the SLT respondents from Austria (0.0%). In Germany, around half of the respondents have less than 50% of multilingual patients on their caseload, while the other half have more than 50% of multilingual patients on their caseload. In Austria, by contrast, about 70% of the SLT respondents have less than 50% of multilingual caseloads. In Switzerland, 60% of the SLT respondents have more than 50% of multilingual caseloads (see Figure 3).

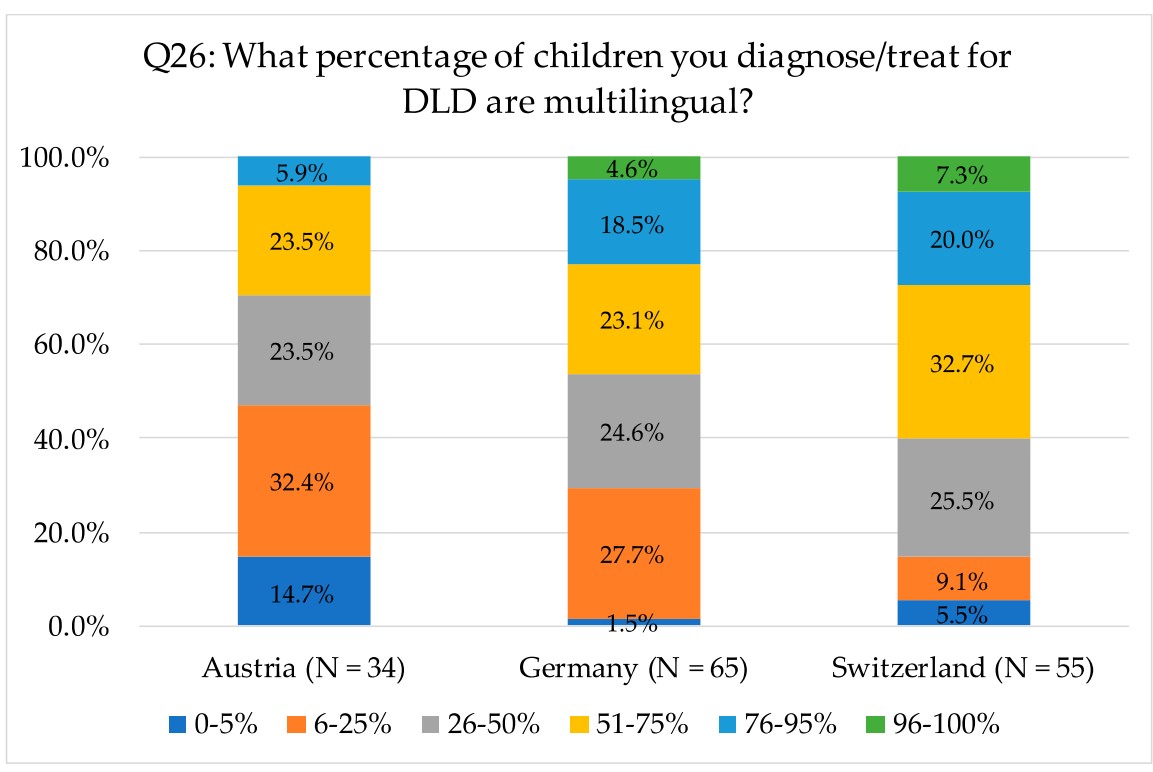

**Figure 3.** Distribution of SLTs' responses concerning the proportion of multilingual children with DLD on their caseloads in the three German-speaking countries.

Table 7 provides an overview of all further questions for which SLTs' responses differed in distribution according to country. We found a significant association of the country with the SLTs' responses concerning beliefs about the impact of multilingualism in the manifestation of DLD (Q4; $X^2$ (9, N = 300) = 20.05, p = 0.018, Cramer's V = 0.15, Phi = 0.26); with their habit to directly observe children to compare their performance in the languages or base their comparisons on the parents' information (Q7; $X^2$ (9, N = 290) = 17.77, p = 0.038, Cramer's V = 0.14, Phi = 0.25); and with the awareness for (Q8; see Figure 4) and the application of testing material for multilingual children (Q9; see Figure 5) (Q8; $X^2$ (3, N = 300) = 32.86, p < 0.001, Cramer's V/Phi = 0.33; Q9; $X^2$ (9, N = 299) = 20.81, p = 0.014, Cramer's V = 0.15, Phi = 0.26). Respondents' attitudes towards usefulness of checking for cross-linguistic influence in morphosyntax (Q12) and the lexicon (Q14, present in the German version of the questionnaire distributed in Austria, Germany, and Switzerland only) also were significantly dependent on country (Q12; $X^2$ $^2$ (9, N = 300) = 18.49, p < 0.030, Cramer's V = 0.25, Phi = 0.14; Q14; $X^2$ (6, n = 197) = 13.31 p = 0.038, Cramer's V = 0.18, Phi = 0.26). Additionally, the SLT respondents' appreciation of the usefulness of homework in the child's heritage language (Q20) and their actual application if they were available (Q21) were significantly different across countries (Q20; $X^2$ (9, N = 299) = 21.40, p = 0.011, Cramer's V = 0.5, Phi = 0.27; Q21; $X^2$ (9, N = 300) = 20.08 p = 0.017, Cramer's V = 0.15, Phi = 0.26). In addition to language-related differences, SLTs' opinions on whether dif-

ferences in the languages spoken by the child and the therapist influence the quality of the intervention (Q22) also varies across countries ($X^2$ (9, $N = 299$) = 21.40, $p = 0.011$, Cramer's $V = 0.15$, $Phi = 0.27$). Finally, the responses concerning the influence of cultural differences on parent consultation (Q24) were significantly modulated by country ($X^2$ (9, $N = 299$) = 19.72 $p = 0.020$, Cramer's $V = 0.15$, $Phi = 0.26$).

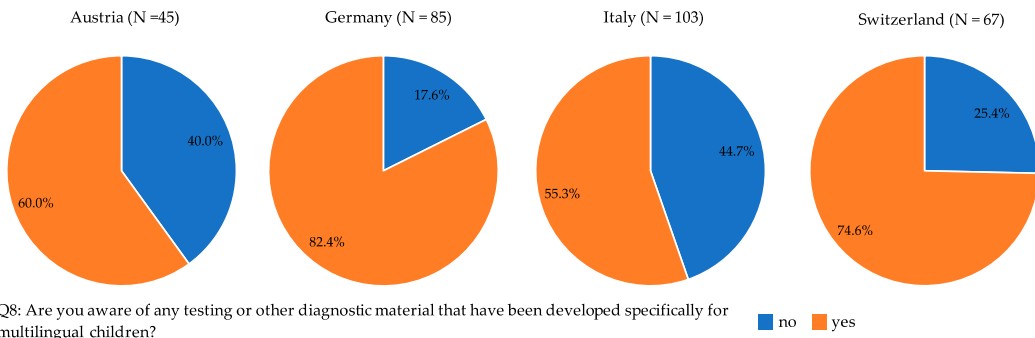

Q8: Are you aware of any testing or other diagnostic material that have been developed specifically for multilingual children?       ■ no ■ yes

**Figure 4.** Distribution of SLTs' awareness of diagnostic material specifically for multilingual children according to country.

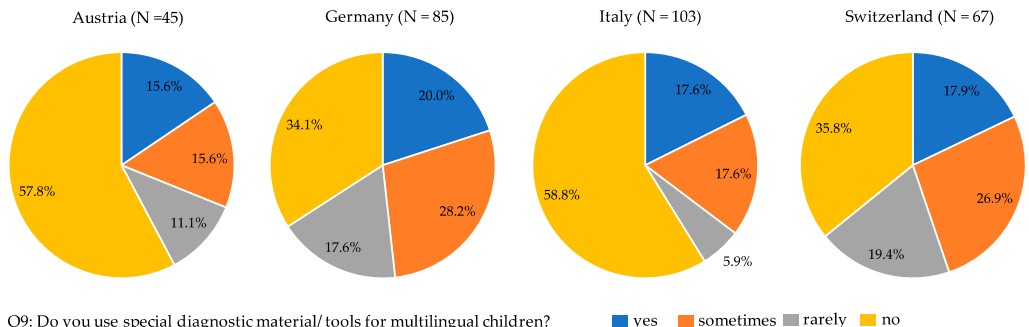

Q9: Do you use special diagnostic material/tools for multilingual children?       ■ yes ■ sometimes ■ rarely ■ no

**Figure 5.** Distribution of SLTs' application of diagnostic material specifically for multilingual children according to country.

**Table 7.** Distribution of SLTs' multilingualism-related attitudes across countries.

| Question | Response Options | Austria | Germany | Italy | Switzerland |
|---|---|---|---|---|---|
| Q4 Do you believe that (A) DLD is independent of speaking a second language, or that (B) multilingualism may impact the manifestation of DLD? | A | 37.8% | 48.2% | 26.2% | 28.4% |
| | more A than B | 17.8% | 20.0% | 23.3% | 35.8% |
| | more B than A | 37.8% | 22.4% | 35.9% | 22.4% |
| | B | 6.7% | 9.4% | 14.6% | 13.4% |
| Q7 if yes/sometimes/rarely (Q6)...(A) were your comparisons based on information provided by the parents or (B) were you able to directly observe child behavior in both languages (possibly in the presence of parents)? | A | 31.1% | 30.6% | 31.1% | 32.8% |
| | more A than B | 26.7% | 41.2% | 26.2% | 43.3% |
| | more B than A | 31.1% | 12.9% | 15.5% | 14.9% |
| | B | 6.7% | 14.1% | 20.4% | 9.0% |
| Q12 Do you think it would be useful to check whether the syntactic and morphological structures that cause difficulties to the child in the societal language are present/also affected in the heritage language? | no | 8.9% | 3.5% | 1.9% | 6.0% |
| | rarely | 6.7% | 2.4% | 0.0% | 0.0% |
| | sometimes | 13.3% | 20.0% | 12.6% | 22.4% |
| | yes | 71.1% | 74.1% | 85.4% | 71.6% |
| Q14 Do you think it would be useful to check whether words that the child uses semantically/lexically incorrectly in German (i.e., the societal language) are also used incorrectly in the heritage language? | no | 4.4% | 2.3% | | 0.0% |
| | rarely | 6.7% | 2.3% | | 3.0% |
| | sometimes | 6.7% | 10.6% | | 25.4% |
| | yes | 82.2% | 84.7% | | 71.6% |
| Q20 Do you think it would be useful to give the child and his/her family tasks to take home to practice in the heritage language? | no | 11.1% | 2.4% | 8.7% | 6.0% |
| | rarely | 4.4% | 10.6% | 1.9% | 14.9% |
| | sometimes | 26.7% | 47.1% | 42.7% | 43.3% |
| | yes | 57.8% | 40.0% | 45.6% | 35.8% |

**Table 7.** *Cont.*

| Question | Response Options | Austria | Germany | Italy | Switzerland |
|---|---|---|---|---|---|
| Q21 If such tasks were available to you (Q20), would you use them? | no | 6.7% | 1.2% | 4.9% | 7.5% |
| | rarely | 2.2% | 7.1% | 1.0% | 13.4% |
| | sometimes | 20.0% | 28.2% | 26.2% | 29.9% |
| | yes | 71.1% | 63.5% | 68.0% | 49.3% |
| Q22 Do you think that differences in the languages spoken by the child and the therapist may influence the quality of the intervention? | no | 8.9% | 18.8% | 5.8% | 7.5% |
| | rarely | 13.3% | 12.9% | 1.0% | 20.9% |
| | sometimes | 42.2% | 44.7% | 48.5% | 55.2% |
| | yes | 35.6% | 23.5% | 44.7% | 16.4% |
| Q24 Do you think that cultural differences between the SLT and the child's family influence the quality of parent consultation? | no | 11.1% | 11.8% | 2.9% | 7.3% |
| | rarely | 11.1% | 11.8% | 1.9% | 8.7% |
| | sometimes | 42.2% | 42.4% | 45.6% | 45.3% |
| | yes | 35.6% | 34.1% | 48.5% | 38.3% |

### 3.5. Effects of Duration of SLTs' Education and Language Background

The potential influence of SLTs' level of education (Q29; "What is your educational qualification?" response options: "Bachelor's degree in SLT (3 years)", "Master's degree in SLT (5 years)", "other") was only taken into account for Italian respondents (*n* = 103). For none of our questions did we find a significant effect for the level of education (*p* > 0.05). We did not find any significant differences between SLTs' attitudes and approaches in working with multilingual children depending on whether or not they spoke more than one language themselves (Q27; "Besides German, do you speak any other language(s) at native level?" and Q28 "What other language(s) do you speak at native level?" ($p_s$ > 0.05), a variable that was only included in the German translation of the questionnaire.

### 3.6. Effects of SLTs' Attitudes towards Children's Heritage Language

We found several significant associations between SLTs' diagnostic and therapeutic practice and recommendation of heritage language use at home (Q5; "Do you think that (A) it would be better for multilingual children with DLD to speak only one language (both at home and outside of the family environment), or that (B) it would be better for all of a child's communication partners to interact with the child in the language they know best?"). These effects are reported in Table 8.

A close-to-significant effect emerged for the association of Q5 with Q3 ($X^2$ (3, *N* = 299) = 7.24, *p* = 0.065, Cramer's *V/Phi* = 0.16). Of the SLTs who support heritage language input at home, only 2.3% exclude the heritage language from the therapeutic setting (Q3). By contrast, among the SLTs who are not supportive of family heritage language usage for multilingual children with DLD, 9.7% exclude the heritage language from the therapeutic setting. Q5 also produced a marginally significant effect on the value assigned by SLTs to the comparison of the multilingual children's different languages (Q6) ($X^2$ (3, *N* = 297) = 7.55, *p* = 0.056, Cramer's *V/Phi* = 0.16). SLTs who support the use of the heritage language at home also think that it is useful to compare performance between all languages spoken by the child (66.0%). By contrast, SLTs who are in favor of limiting children's language input to the societal language tend to think (44.8%) that it is useful to compare a child's performance between the societal and the heritage language.

A significant association was found between Q5 and Q4, i.e., the belief that DLD is completely independent of the languages spoken or rather that multilingualism may impact the manifestation of DLD ($X^2$ (3, *N* = 300) = 21.62, *p* < 0.001, Cramer's *V* = 0.27, *Phi* = 0.26). This effect is probably best interpreted in the reversed direction: The therapists who clearly believe that DLD is independent of multilingualism significantly more often recommend that each family member should speak the language they know best with the child (97.1%), as opposed to the SLTs who believe that multilingualism is not fully independent of DLD, of which only 71.4% recommended the use of the heritage language at home (see Figure 6). Nonetheless, as shown in the figure, even among the SLTs who

support heritage language use at home, 25.3% do think that multilingualism may impact the manifestation of DLD (Q4).

Furthermore, Q5 produced a significant effect on Q8 ($X^2$ (1, $N$ = 300) = 9.89, $p$ = 0.002, Cramer's *V/Phi* = 0.18): SLTs who favor the use of the heritage language are significantly more aware of diagnostic material specifically developed for multilingual children (67.3%) than SLTs who are not supportive of the use of the heritage language at home (38.7%). An association was found also with interest in computerized assessments (Q18) ($X^2$ (3, $N$ = 300) = 10.67, $p$ = 0.014 Cramer's *V/Phi* = 0.19). Of the SLTs who do not think it would be useful to have computerized tasks to assess the level of proficiency in the child's heritage language (5.3% of the total sample), 16.1% were in favor of reducing a child's language input at home to the societal language.

**Table 8.** Distribution of SLTs' attitudes towards heritage language use and approaches in their SLT service provision.

| | | Q5 Do You Think That . . . | |
|---|---|---|---|
| | | **. . . It Would be Better for Multilingual Children with DLD to Speak Only One Language (Both at Home and Outside the Family Environment).** | **. . . It Would be Better for All of a Child's Communication Partners to Interact in the Language They Know Best.** |
| Q3 Do you think that (A) the therapy of DLD in multilingual children should be exclusively limited to the societal language, or that (B) the child's heritage language should also be taken into account? | A | 9.7% | 2.2% |
| | more A than B | 35.5% | 36.9% |
| | more B than A | 22.6% | 36.2% |
| | B | 33.3% | 24.6% |
| Q6 In the context of SLT for multilingual children with DLD, do you think that it is useful to compare a child's language performance in the heritage and societal language? | no | 6.9% | 1.5% |
| | rarely | 6.9% | 4.5% |
| | sometimes | 41.4% | 28.0% |
| | yes | 44.8% | 66.0% |
| Q8 Are you aware of any testing or other diagnostic material that have been developed specifically for multilingual children? | no | 61.3% | 32.7% |
| | yes | 38.7% | 67.3% |
| Q18 Do you think that computerized tasks to assess the level of proficiency in the child's heritage language would be useful? | no | 16.1% | 4.9% |
| | rarely | 9.7% | 4.8% |
| | sometimes | 19.4% | 34.2% |
| | yes | 54.8% | 56.9% |

The SLTs who clearly believe that DLD is independent of multilingualism significantly more often recommend that each family member should speak the language they know best with the child (97.1%), as opposed to the SLTs who believe that multilingualism is not fully independent of DLD, of which only 71.4% recommended the use of the heritage language at home ($X^2$ (3, $N$ = 300) = 21.62, $p$ < 0.001, Cramer's $V$ = 0.27, *Phi* = 0.26) (see Figure 6). Nonetheless, even among the SLTs who support heritage language use at home, 25.28% do think that factors related to multilingualism may aggravate DLD (Q4). In fact, among all respondents, 29.3% tend to think and 11.7% think that multilingual language acquisition plays a role in the manifestation of DLD (see Figure 6).

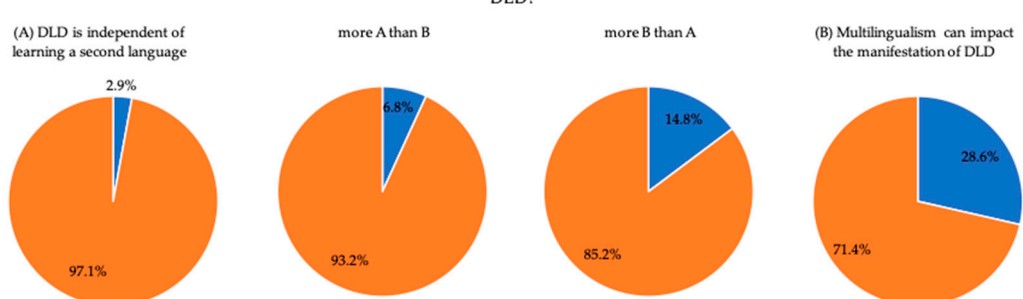

**Figure 6.** Distribution of SLTs' attitudes towards heritage language use subdivided according to their belief about the association between multilingualism and DLD.

## 4. Discussion

The results of the survey partly confirm the research hypotheses. As expected, the large majority of SLT respondents does acknowledge the importance of taking into account aspects of the children's heritage language(s) and culture(s) when treating them and providing parent consultation (Hypothesis 1). Additionally, the hypothesis that SLTs would express interest in, and willingness to use, specialized tools that facilitate SLT service provision for multilingual children was partly confirmed (Hypothesis 2). More surprising was the finding that, while most SLT respondents did state to know about such materials, the majority of respondents did not apply them on a regular basis in their clinical practice. Our results confirmed the hypothesis that differences among countries would emerge in terms of experience and caseload in working with multilingual children (Hypothesis 3), which might be the consequence of both historical (especially migration history) and organizational (regarding the referral and cost coverage for SLT services in the respective health care system) differences. The hypothesis that SLTs' experience in working with multilingual children would influence awareness of, as well as knowledge of, multilingual approaches was also confirmed by our findings (Hypothesis 4). As opposed to Hypothesis 5, younger and thus less experienced SLTs (indicated by their overall work experience in years) were more open towards multilingually oriented approaches in SLT practice. By contrast, the duration of SLT education or training did not show any effect on the attitudes and approaches towards multilingualism in SLT service provision (Hypotheses 6). Neither our hypothesis of an effect of the SLTs' own language background on their attitudes and approaches (Hypothesis 7) nor the hypothesis that knowledge and usage of special material for diagnosing and treating multilingual children would be strictly related (Hypothesis 8) were confirmed by our data. Finally, our hypothesis that a positive attitude towards the use of the heritage language at home would be associated with openness to multilingual and multimodal approaches in SLT service provision was confirmed (Hypothesis 10).

These results can be further discussed by highlighting a series of crucial factors that were found to influence SLTs' attitudes and practice with respect to multilingualism.

### 4.1. Effects of Experience and Education

Overall experience in years, which is most likely linked to the recency of participants' training, was found to be influential for a subset of questions. SLTs who graduated more recently seem to be more open to multilingual approaches. This may be an indication that in recent years the topic of multilingualism has received more attention in SLT training. However, this general professional experience in years of our respondents turned out to have little influence on their attitudes and approaches in SLT service for multilingual children. We found experience in specifically working with multilingual children to be the most influential factor on SLTs' attitudes and approaches towards multilingualism. This

suggests that SLT education should provide appropriate information and practical training on the topic of multilingualism. Since we found no significant effects of length of education on attitudes and approaches in SLT service for multilingual children, it seems crucial to not only provide theoretical knowledge but also concrete field practice in assessing and treating multilingual children

An interesting effect that was highlighted by the results when analyzing the influence of experience is that of polarization of attitudes. While the least experienced SLTs showed a preference for the more polarized/extreme attitudes, SLTs with the highest level of experience tended to choose less clear-cut answers. This may suggest that with increasing experience SLTs become more able to understand and acknowledge the heterogeneity of their multilingual patients and the complexity of the factors involved in each individual situation and are thus more likely to take each family's highly individual language and cultural background into account.

### 4.2. Country-Related Differences: Historical, Social, and Health Policy-Related Factors

Country-specific differences in SLTs' experience in working with multilingual children are likely to reflect the specific patterns of the history of immigration in each of the four countries (i.e., least experienced SLTs in Italy where migration is more recent). A more detailed analysis of SLTs' multilingual caseloads across the three German-speaking countries provided more fine-grained insights into SLTs' overall contact with multilingual children. Despite the fact that for many SLTs multilingualism is a regular part of their everyday practice, for some of them the topic of multilingualism is still a peripheral part of their work and the number of SLTs who have specialized multilingual caseloads (more than 96% of their patients) is relatively low. Previous studies have found barriers for migrant families' access to healthcare services (and, hence, also SLT services), which differ from the majority population [17]. This inequality is also reflected in our data. The proportion of multilingual children on SLTs' caseloads was not in accordance with the percentage of people with migration background across the three German-speaking countries. In Austria, we saw fewer multilingual caseloads despite the same proportion and origin of migrants compared to Germany. It may be noteworthy that a considerable proportion of Austrian immigrants migrate from Germany (13.6%; see Table 2). These children, in particular, may not have been considered multilingual by SLT respondents from Austria. However, this alone cannot account for the differences in SLTs' multilingual caseloads, and it is plausible that the effect we found is reflective of the differences in the referral process and therapy cost coverage between the two countries.

Lower awareness and application of special diagnostic materials in Italy might be related to SLTs' lesser experience in working with multilingual children. which again is related to the smaller rate of the foreign-born population. Additionally, due to the fact that immigration is a rather recent phenomenon in Italy compared to the other three countries, the need for such material is even more recent. Indeed, Italian SLTs were less experienced and more polarized in their responses when judging the usefulness to compare (morphosyntactic structures in) the languages spoken by the child. While in Italy, as previously stated, the identification of a need for such material may have arisen relatively recently, in Germany (1) immigration is more common/greater in number and (2) the academic environment for SLTs and thus development of multilingual material are relatively greater. SLTs in Germany—and, due to similarities in terms of language, also Austria and Switzerland—may thus have easier access to diagnostic material for multilingual children. This easier access to diagnostic material could, in fact, facilitate the process of language comparison and influence the respondents' perception of the material's usefulness. Moreover, based on more practical experience, SLTs could also be more aware of the additional effort needed when applying special diagnostic material for multilingual children and of the importance to weigh such effort compared to the benefits deriving from its use, case by case.

Since no significant difference emerged between SLTs' attitudes towards heritage language across countries, we interpret the country-related differences in SLTs' estimates of usefulness and application of homework in the child's heritage language as reflecting a different role and representation of the value of homework in the culture of the respective countries. We especially found that Swiss SLTs were least likely to use homework in the heritage language of the child, which might be related to the fact that, in Switzerland, many children receive SLT at school where the contact between SLTs and parents who support the children during homework activities might be limited.

The significant difference regarding the evaluation of the impact of multilingualism on intervention quality may be due to, as we stated, translation differences in the two language versions of the questionnaire ("the fact that the therapist and the child speak different languages may affect the quality of the intervention" in the German version; "linguistic differences between the language spoken by the child and the therapist may affect the quality of the intervention" in the Italian version). However, the significant, country-specific differences regarding SLTs' estimates that the quality of parent consultation is influenced by differences between a family's and the SLT's culture cannot be traced back to discrepancies between the two language versions. The fact that Italian SLT respondents most often agreed with the statement that cultural differences may influence parent consultation appears not to depend on their reduced level of experience and might instead reflect a greater apprehension towards the difficulties involved in intercultural contacts, also due to the relatively recent migration history in Italy.

### 4.3. SLTs' Language Background

Surprisingly, we did not find an effect of the SLTs' own language background on their attitudes and approaches towards SLT service provision for multilingual children. One way to interpret this finding is that the lack of adequate material has a more severe impact than the lack of a multilingual background. However, it should also be considered that half of the multilingual SLT respondents in our sample spoke a language that is not represented in the heritage languages most common among the migration groups of the respective country. We propose two potential explanations for this finding: (1) A great proportion of the multilingual SLTs within our sample may not be able to use their own language skills in the context of SLT and thus adhere to more monolingual-like practices and (2) the SLTs' personal language status (being monolingual vs. multilingual), which we assumed to be associated with the habit of reflecting about cross-linguistic phenomena and interaction between different languages, turned out to not increase their awareness of the importance to examine children's language performances in their various languages. Therefore, being multilingual does not constitute an advantage to adequately assess and treat multilingual children or to appreciate the complexity and the needs related to multilingualism. Nevertheless, it can be advantageous in that it enables SLTs to provide service in more than one language for multilingual children.

### 4.4. Discrepancies between Knowledge and Practice

We found widespread awareness for multilingual approaches and multilingually oriented attitudes among SLT respondents. The majority is open to learning about and to using new and suitable material and techniques. However, several hurdles make the actual application of such material and techniques difficult. We offer a two-fold explanation for this discrepancy between knowledge and practice, namely, that the material is insufficiently usable because (1) it is only applicable for a small number of children on a therapist's caseload and/or not affordable and (2) it requires the therapist's manual evaluation of a child's responses, which presumes that the therapist has already acquired a certain amount of knowledge about the child's heritage language, which is very time-consuming. The gap between the high knowledge and low usage of specialized multilingual material might thus be explained by the high demand on SLTs' resources (time and material) that would be required for its application.

We found another very specific discrepancy between SLTs' beliefs about the impact of multilingualism in the genesis of DLD and their practice. Although some participants believed that factors related to multilingualism may exacerbate DLD, they nevertheless were supportive of heritage language usage at home. This finding is likely to originate from the non-univocal interpretation of the question, where the statement "DLD may be exacerbated by factors related to multilingualism" might evoke both an outdated view about detrimental effects of multilingualism on language development and also the valuable awareness of the effects that cross-linguistic influence may exert on language functions. Indeed, it might also reflect a certain confusion between the effects of multilingualism and the effects of low SES. This is particularly important in the light of Roy and Chiat's [10] statement that a certain proportion of low SES (i.e., often migrant and thus multilingual) children who, based on their scores in standardized language assessment tools, supposedly present with DLD have "intact language potential". The authors further argue that if these children had only been exposed to a more advantageous environment, they would have developed language skills within the normal range. That being said, these children still require and are entitled to adequate language support in order to counteract the effect of accumulated disadvantage. In an additional and alternative perspective, Stankova et al. [50] suggest that the discrepancy between clinical practice and SLTs' common beliefs/attitudes towards multilingualism among SLTs reflects the fact that most of the countries sampled in previous research have been historically monolingual and that there is thus a latency in the clear appreciation of the need for professional linguistic competence in key languages. High-quality and multilingually adequate assessment and/or therapy require more time and resources on the side of the therapist as he/she needs to learn about language-specific linguistic phenomena and their sequence of acquisition for each of his/her patient's heritage language.

### 4.5. Limitations and Future Perspectives

One limitation of this study is that, based on a priori expectations, some questions were either only asked in the Italian- or the German-speaking sample. Participant demographics could have been investigated in more detail in order to allow for a more in-depth interpretation of the results (demographic data should have included participants' gender, age, and country of origin). Moreover, more efforts could have been devoted to avoiding translation-related differences in multilingual versions of the questionnaires.

To provide a more complete picture of the differences among and within countries, future studies should compare language support services and SLT services in terms of their accessibility, costs and cost coverage, and success rate. Additional information could come from investigating whether and how well SLTs speak a more typical minority language. It would also be interesting to assess the usability of already existing tools aiming at disentangling DLD from imbalanced multilingual language acquisition and to look into the question of why they are not applied in clinical practice.

### 4.6. Practical Implications

The barriers to adequately assessing and treating multilingual children with DLD seem to be multifactorial in nature and the limited access to several key supports and resources for overcoming some of these barriers (i.e., easy access to interpreters or assessment tools in languages beyond the societal language) might explain why the vast majority of SLTs associate the assessment of multilingual children with greater effort [34], which also helps to maintain the current predominance of monolingual SLT policies and practices.

The results suggest that more preparation time may be needed by SLTs when working with multilingual children, and that this should be acknowledged and financially compensated by the health care providers.

A further reflection concerns entrance exams for SLT programs, in line with Scharff Rethfeldt's [17] notion of monolingual and -cultural focus in SLT practice. In most countries, entrance exams for SLT training focus on language performance in the societal language

and applicants' ability to speak another language is hardly taken into account in the process of accepting future SLT students. In order to meet the needs of migrant populations and provide SLT services particularly in the heritage languages that are most common among a country's residents, we need to move beyond the current predominance of monolingual and -cultural practices. Facilitated access to SLT training for second-language speakers of the societal language, minority language classes as part of the SLT curriculum, or re-training for people with migration backgrounds who already have a professional background in SLT-related areas could constitute a first step towards this goal. The issue of multilingualism in SLTs' education and studies should be made an explicit subject of discussion. This does not only include theoretical knowledge on multilingual language acquisition and the presentation of adequate assessment and intervention material, but also sensitization to intercultural competence and communication. Considering the need for further development of knowledge and of ad hoc instruments in the field of multilingualism and its impact on DLD, the involvement of SLTs in research should also be encouraged and supported.

Finally, the finding that practical experience seems to be the most influential factor on SLTs' attitudes and approaches towards multilingualism points to the need to ensure that concrete experience with multilingual patients is firmly established in the SLTs' training curriculum. Furthermore, more opportunities for continuing education for in-service SLTs could further raise awareness for evidence-based multilingual approaches in SLT service provision. In order to support in-service SLTs to pursue continuing education, time and costs for participating in such classes should be compensated by the health care system.

## 5. Conclusions

Our data suggest that neither duration of education nor duration of overall professional experience but experience in SLT service provision for multilingual children is associated with the respective attitudes and approaches towards multilingualism in SLT.

In spite of the SDG 10 "Reduce inequalities" [3], there is a risk for residual inequalities in the provision of SLT to multilingual children that should be addressed. The additional efforts required to adequately address multilingualism-related issues should not lead to more inequality in the provision of SLT services but requires substantial infrastructural and systemic changes.

The complexity related to multilingualism can only be met through increased resources and multi-professional awareness and knowledge (not only in SLTs but also in pediatricians who are responsible for children's referrals, teachers, and other professionals working with children) in order to provide adequate diagnosis and intervention not ignoring but rather understanding and correctly addressing all the mechanisms involved in the parallel development of different languages in a synergic perspective.

**Author Contributions:** T.B. and M.E. share first authorship. All co-authors contributed to the manuscript. Conceptualization, M.L.L. and M.E.; Methodology, M.L.L.; Online questionnaire implementation, M.L.L. and T.B.; Formal Analysis, M.L.L.; Investigation, T.B., M.E., and M.L.L.; Resources, T.B., M.E., T.R. and M.L.L.; Data Curation, T.B., M.E. and M.L.L.; Writing—Original Draft Preparation, T.B. and M.E.; Writing—Review and Editing, T.B., M.E., T.R. and M.L.L.; Visualization, T.B. and M.E.; Supervision, M.L.L. and T.R.; Project Administration, M.L.L. and T.R.; Funding Acquisition, M.L.L. and T.R. All authors have read and agreed to the published version of the manuscript.

**Funding:** This project received funding from the European Union's Horizon 2020 program for research and innovation under the Marie Skłodowska Curie Grant Agreement No. 765556 and by the Italian Ministry of Health, Grant RC2021 to Maria Luisa Lorusso. Article Processing Charges were supported by the German Research Foundation (DFG) within the funding program Open Access Publishing via the Catholic University Eichstätt-Ingolstadt.

**Institutional Review Board Statement:** The general framework of the MultiMind project was approved by the Ethics Committee of the Scientific Institute Eugenio Medea, scientific section of the association "La Nostra Famiglia" Prot. N. 43/19, 17 June 2019. With regard to this specific survey ethi-

cal review and approval were waived, since the study was conducted with a group of non-vulnerable healthy adults in a completely anonymous form.

**Informed Consent Statement:** The data described were collected in a fully anonymous online survey. Prior to their participation, all participants were informed that their anonymous responses would be used for scientific purposes, and that by filling out the questionnaire they would accept these conditions.

**Data Availability Statement:** This research is not based on publicly archived data sets analyzed or generated during the study. The data presented in this study are openly available in Zenodo at https://doi.org/10.5281/zenodo.5524401 (accessed on 23 September 2021).

**Acknowledgments:** We would like to thank the various professional associations in the different countries for collaboration in advertising the survey among their members: in Italy, the Italian Federation of Speech-and-Language Therapists (FLI); in Austria, logopädieaustria; in Germany, the Bundesverband für Logopädie e.V. (dbl) as well as the Deutsche Bundesverband für akademische Sprachtherapie und Logopädie (dbs); and in Switzerland, the Deutschschweizer Logopädinnen- und Logopädenverband (DLV). Furthermore, we would like to thank all anonymous participants who took part in the survey as well as those involved in the piloting phase for their helpful feedback.

**Conflicts of Interest:** The authors declare no conflict of interest. The funding parties had no role in the design of the study; in the collection, analyses, or interpretation of data; in the writing of the manuscript; or in the decision to publish the results.

## Appendix A

**Table A1.** Overview of the questions.

| Question Number | Question | Answering Options |
|:---:|:---:|:---:|
| Q1 | Have you ever provided therapy to multilingual children with developmental language disorder (DLD)? | Yes<br>Sometimes<br>never |
| Q2 | Do you think that different approaches are needed in the diagnosis and treatment of multilingual children compared to monolingual children? | Yes<br>Sometimes<br>never |
| Q3 | Do you think that (A) the therapy of DLD of multilingual children should be limited exclusively to the societal language, or that (B) the child's first language should also be taken into account? | A<br>more A than B<br>more B than A<br>B |
| Q4 | Do you believe that (A) DLD is independent of speaking a second language, or that (B) multilingualism can impact the manifestation of DLD? [2] | A<br>more A than B<br>more B than A<br>B |
| Q5 | Do you think that (A) it would be better for multilingual children with DLD to speak only one language (both at home and outside of the family environment), or that (B) it would be better for all of a child's communication partners to interact with the child in the language they know best? | A<br>B |
| Q6 | In the context of speech and language therapy (SLT) for multilingual children with DLD, do you think that it is useful to compare a child's language performance in his/her first and second language? | Yes<br>Sometimes<br>Rarely<br>never |

**Table A1.** *Cont.*

| Question Number | Question | Answering Options |
|---|---|---|
| Q7 | If yes/sometimes/rarely (Q6) . . . (A) were your comparisons based on information provided by the parents or (B) were you able to directly observe child behavior in both languages (possibly in the presence of parents)? | A<br>more A than B<br>more B than A<br>B |
| Q8 | Are you aware of any testing or other diagnostic material that have been developed specifically for multilingual children with DLD? | Yes<br>no |
| Q9 | Do you use special diagnostic material/tools for multilingual children with DLD? | Yes<br>Sometimes<br>Rarely<br>never |
| Q10 | If yes/sometimes/rarely (Q9) . . . What diagnostic material/tools do you use for multilingual children with DLD? | [open question, free text answer] |
| Q11 | Do you think it would be useful to check whether the phonemes that cause difficulties to the child in the societal language are present [3]/also affected [4] in the heritage language? | Yes<br>Sometimes<br>Rarely<br>never |
| Q12 | Do you think it would be useful to check whether the syntactic and morphological structures that cause difficulties to the child the societal language are present [5]/also affected [6] in the heritage language? | Yes<br>Sometimes<br>Rarely<br>never |
| Q13 | Do you think it would be useful to check whether words that the child uses semantically/lexically incorrectly in the societal language are similar or very different in the heritage language? | Yes<br>Sometimes<br>rarely<br>never |
| Q14 | Do you think it would be useful to check whether words that the child uses semantically/lexically incorrectly in German (i.e., the societal language) are also used incorrectly in the heritage language? [7] | Yes<br>Sometimes<br>Rarely<br>never |
| Q15 | Do you think it would be useful to have a summary chart of the phoneme inventory of the child's heritage language? | Yes<br>Sometimes<br>Rarely<br>never |
| Q16 | Do you think it would be useful to have a summary table of the main syntactic structures and constructions in the child's heritage language? | Yes<br>Sometimes<br>Rarely<br>never |
| Q17 | Do you think that an overview table with a list of the most important prepositions (translation + usage) in the child's heritage language would be useful? | Yes<br>Sometimes<br>Rarely<br>never |
| Q18 | Do you think that computerized tasks to assess the level of proficiency in the child's other language would be useful? | Yes<br>Sometimes<br>Rarely<br>never |
| Q19 | If such tasks existed, would you use them when working with multilingual children with DLD? | Yes<br>Sometimes<br>Rarely<br>never |

**Table A1.** *Cont.*

| Question Number | Question | Answering Options |
|---|---|---|
| Q20 | Do you think it would be useful to give the child and his/her family tasks to take home to practice in heritage language? | Yes<br>Sometimes<br>Rarely<br>never |
| Q21 | If such tasks were available to you, would you use them? | Yes<br>Sometime<br>rarely<br>never |
| Q22 | Do you think that differences in the languages spoken by the child and the therapist may influence the quality of the intervention? | Yes<br>Sometimes<br>Rarely<br>never |
| Q23 | Do you think that differences in the languages spoken may the quality of the parent consultation? | Yes<br>Sometimes<br>Rarely<br>never |
| Q24 | Do you think that cultural differences between the SLT and the child's family influence the quality of parent consultation? | Yes<br>Sometimes<br>Rarely<br>never |
| Q25 | For how many years have you practiced/have you been practicing Speech and Language Therapy? | <5 years<br>5–10 years<br>10–20 years<br>>20 years |
| Q26 | What percentages of children you diagnose/treat for DLD are multilingual? | 0–5%<br>6–2%<br>26–50%<br>51–75%<br>76–95%<br>96–100% |
| Q27 | Besides German, do you speak any other language(s) at native level? [8] | Yes<br>no |
| Q28 | If yes (Q27), what language(s) do you speak any other language(s) at native level? [9] | [open question, free text answer] |
| Q 29 | What is your educational qualification? [10] | BA in SLT (3 years)<br>MA in SLT (5 years)<br>other |

[2] Language versions varied in their precise formulation: While the Italian translation specifically asked about effects of cross-linguistic influence (Do you think that (A) if the disorder is specific then it is independent of the heritage language or (B) even if the disorder is specific it can be aggravated by influence from the heritage language?), the German question targeted the general impact of multilingualism (Do you think that (A) DLD is independent of the second language, or that (B) a disorder can be exacerbated by second language acquisition?). [3] Italian version of Q11 (used in the Italian version of the questionnaire spread in Italy). [4] German version of Q11 (used in the German version of the questionnaire spread in Austria, Germany, and Switzerland). [5] Italian version of Q11 (used in the Italian version of the questionnaire spread in Italy). [6] German version of Q11 (used in the German version of the questionnaire spread in Austria, Germany, and Switzerland). [7] This question is present in the German version only. [8] This question is present in the German version only. [9] This question is present in the German version only. [10] This question is present in the Italian version only.

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
