# Peer review of "Speech and Language Therapy Service for Multilingual Children: Attitudes and Approaches across Four European Countries"

_sustainability, doi:10.3390/su132112143_

Round 1

Reviewer 1 Report

A thorough and valuable study. The authors should be praised for the choice of their research topic.

This research contributes to better understanding of the rapidly changing social reality of increasingly multicultural societies resulting from migration processes. It also has valuable practical implications for the community of SLTs and policy makers in the four countries.

Background information on ethnic context of participating countries is fully provided, research procedures and results are clearly presented.

Use of English is excellent, I only noted very few small errors  (see lines 620, 622, 647).

I have one suggestion to the authors - perhaps in view of their findings, in the discussion (especially in the section “Practical implications”) the issue of provision of in-service training for SLTs could be raised? It would compensate for what they suggest is inadequate training about multilingualism during their preparatory studies (p. 6 in the article).

Author Response

Dear Reviewer 1,

We have incorporated your suggestions. Those changes are highlighted within the manuscript. Please see below, in bold, for a point-by-point response to your comments and concerns. All line and section numbers refer to the revised manuscript file with tracked changes.

  1. Comment by Reviewer 1

A thorough and valuable study. The authors should be praised for the choice of their research topic
This research contributes to better understanding the rapidly changing social reality of increasingly multicultural societies resulting from migration processes. It also has valuable practical implications for the community of SLTs and policy makers in the four countries.
Background information on ethic context of participating countries is fully provided, research procedures are results are clearly presented.

Thank you very much for your positive feedback. It is encouraging to hear that our research topic is considered relevant.

  1. Comment by Reviewer 1

Use of English is excellent, I only noted a very few small errors (see lines 620, 622, 647).

Thank you for pointing out our errors which we hope to have corrected appropriately (see lines 719, 721, 744).

  1. Comment by Reviewer 1

I have one suggestion to the authors – perhaps in view of their findings, in the discussion (especially in the section “Practical implications”) the issue of provision of in-service training for SLTs could be raised? It would compensate for what they suggest is inadequate training about multilingualism during their preparatory studies (p. 6 in article)

Thank you for your valuable suggestion concerning the notion of in-service training for SLTs in the section on practical implications (4.6) that we have added.

We’d like to thank you for your insightful suggestions, which have – we think and hope – been helpful for improving the manuscript.

Thank you for your attention,

Theresa Bloder, on behalf of all authors

Reviewer 2 Report

The paper examines a topic that is extremely relevant from a scientific point of view and for its practical and clinical implications. It analyses the beliefs of speech therapists regarding various techniques and methodologies for the assessment and intervention in language disorders in children growing up in a multilingual environment. The main finding of the study suggests how important it is to train speech therapists by providing specific training regarding intervention in language disorders in multilingual children.

 The study is well thought, adequately conducted and the results are correctly interpreted. However, at the moment the paper needs some revisions in order to achieve its full potential. Below I try to give some suggestions that the authors might find useful in order to improve the quality of the paper.

 Introduction. In general, the introduction is very well written and the literature is accurately cited and reported. In paragraph 1.3 it is suggested to be more explicit when referring to risks of difficulties due to migrant background or SES. Several recent studies show that these two factors are independent and act independently. therefore, it is relevant to consider them separately and make explicit when referring to one and when referring to the other.

In the same paragraph, it is generically stated that the speech therapists consider more effective the evaluation and the treatment in both languages spoken by the child. it would be relevant to report the scientific literature that confirms a greater effectiveness of this type of treatments compared to treatments concerning L1 or L2. To my knowledge, there are not many studies reporting on the effectiveness of treatment in both languages. generally, the treatment takes place in the language of the society and therefore the data on effectiveness generally refer to this.

In section 1.3.1 several studies are reported where the importance of L1 assessment in multilingual children is illustrated. a number of studies are dismissed in this section which instead demonstrate the effectiveness of adequate assessment for diagnostic purposes when assessing L2 alone and only indirectly L1.

In section 1.3.2 with respect to the absence of standards for multilingual children, it is considered relevant that a brief discussion is added regarding the fact that literacy for bilingual children generally takes place in the L2. therefore the assessment of L2 using the norms of monolinguals is relevant in order to establish the extent to which the bilingual child falls below compared to monolingual peers. This is informative with respect to his/her school readiness.

It is suggested to the authors to consider citing the works of Bonifacci and those of Paradis that offer a relevant overview with respect to the most reliable methodologies for the assessment of bilingual children even in the absence of the possibility of assessing them in the L1. in fact in these studies they suggest which tests to use for the diagnosis of the language disorder in the bilingual child.

 Finally, it is suggested that the authors consider citing the Cost Action IS0804. and the Litmus network - a research group in which many assessment tests appropriate for multilingual children are offered for the diagnosis of language disorder in this population.

method. in my opinion there should be a Stronger rationale to justify the presence of the 4 countries involved in the study. i am not convinced that there is a theoretical clinical or cultural reason to compare exactly these four countries. therefore in my opinion the authors should limit describe the situation in the four countries.  Alternatively, it would be relevant to elaborate on what exactly comparing the different countries tells us.

 results.  For the sake of clarity the authors might give subtitles to the various paragraphs so that the reader can follow each objective and the corresponding result more easily.

 in general, the results would need more structure.

 my biggest concern about the data analysis is that the study was conducted in the 4 countries but then all the data were put together. what justifies this.? are the performances similar? Then there are analyses comparing the 4 countries. What does this indicate (see also my point about the method)?

Finally, I would like to invite the authors to consider the possibility of conducting analyses correlating the answers to the questionnaire and the personal characteristics of the participants, such as age, gender, type of work (public/private), country of origin. in my opinion, this type of analysis could further enrich the very interesting results of this study and could favour some interpretations that the authors have already provided.

 The discussion should be revised on the basis of the changes made in the other sections.

Author Response

Dear Reviewer 2,

We have incorporated most your suggestions. Those changes are highlighted within the manuscript. Please see below, in bold, for a point-by-point response your comments and concerns. All section numbers refer to the revised manuscript file with tracked changes.

In the introduction, the parts on SES were revised and a more thorough and specific context on its relationship with DLD has been provided. Furthermore, we have explained better why we didn’t emphasize the analysis comparing the four countries.

  1. Comment by Reviewer 2

The paper examines a topic that is extremely relevant from a scientific point of view and for its practical and clinical implications. It analyses the beliefs of speech therapists regarding various techniques and methodologies for the assessment and intervention in language disorders in children growing up in a multilingual environment. The main finding of the study suggests how important it is to train speech therapists by providing specific training regarding intervention in language disorders in multilingual children.

The study is well thought, adequately conducted and the results are correctly interpreted. However, at the moment the paper needs some revisions in order to achieve its full potential. Below I try to give some suggestions that the authors might find useful in order to improve the quality of the paper

The study is well thought, adequately conducted and the results are correctly interpreted. However, at the moment the paper needs some revisions in order to achieve its full potential. Below I try to give some suggestions that the authors might find useful in order to improve the quality of the paper

Thank you very much for your positive feedback. It is encouraging to hear that our research topic is considered relevant.

Thank you for your support in improving our manuscript.

  1. Comment by Reviewer 2

Introduction. In general, the introduction is very well written and the literature is accurately cited and reported. In paragraph 1.3 it is suggested to be more explicit when referring to risks of difficulties due to migrant background or SES. Several recent studies show that these two factors are independent and act independently. therefore, it is relevant to consider them separately and make explicit when referring to one and when referring to the other.

Thank you for this comment. We have strengthened the part on SES and DLD in section 1.1 and elaborated the issue of SES and migration background in section 1.3. We hope it is clearer now.

  1. Comment by Reviewer 2

In the same paragraph, it is generically stated that the speech therapists consider more effective the evaluation and the treatment in both languages spoken by the child. it would be relevant to report the scientific literature that confirms a greater effectiveness of this type of treatments compared to treatments concerning L1 or L2. To my knowledge, there are not many studies reporting on the effectiveness of treatment in both languages. generally, the treatment takes place in the language of the society and therefore the data on effectiveness generally refer to this

Thank you for raising this interesting point. In section 1.3 we have now cited studies by Thordadottir and colleagues (2015) and Restrepo, Morgan, and Thomson, (2013) that have shown the limited transfer of the effects of SLT delivered in only the L2 to the L1. In line with others, they highlight the relevance of providing a bilingual context during the intervention in order for the child to be able to make improvements in his/ her language(s) used within and outside the family context.

  1. Comment by Reviewer 2

In section 1.3.1 several studies are reported where the importance of L1 assessment in multilingual children is illustrated. a number of studies are dismissed in this section which instead demonstrate the effectiveness of adequate assessment for diagnostic purposes when assessing L2 alone and only indirectly L1.

Thank you for highlighting this relevant aspect. We have added several studies on language-universal and L2 assessment material for bilingual children (e.g. Boerma & Blom, 2017; Schulz & Tracy, 2011; LITMUS COST Action IS0804; Bonifacci et al., 2020; Paradis, Schneider, & Duncan, 2013). (Section 1.3.1)

  1. Comment by Reviewer 2

In section 1.3.2 with respect to the absence of standards for multilingual children, it is considered relevant that a brief discussion is added regarding the fact that literacy for bilingual children generally takes place in the L2. therefore the assessment of L2 using the norms of monolinguals is relevant in order to establish the extent to which the bilingual child falls below compared to monolingual peers. This is informative with respect to his/her school readiness.

We agree that children’s competences in the L2 are of great importance especially for their academic development. We have included information on the relevance of L2 assessment and intervention in order to support school-readiness in bilingual children (Section 1.3.2).

  1. Comment by Reviewer 2

It is suggested to the authors to consider citing the works of Bonifacci and those of Paradis that offer a relevant overview with respect to the most reliable methodologies for the assessment of bilingual children even in the absence of the possibility of assessing them in the L1. in fact in these studies they suggest which tests to use for the diagnosis of the language disorder in the bilingual child.

Finally, it is suggested that the authors consider citing the Cost Action IS0804. and the Litmus network - a research group in which many assessment tests appropriate for multilingual children are offered for the diagnosis of language disorder in this population.

As stated above, we have added a part on assessment tools providing tests in L2 and questionnaires addressing L1 competence, also mentioning work by Bonifacci and Paradis. Furthermore, we have briefly described the tools designed by the Litmus network (Section 1.3.1).

  1. Comment by Reviewer 2

method. in my opinion there should be a Stronger rationale to justify the presence of the 4 countries involved in the study. i am not convinced that there is a theoretical clinical or cultural reason to compare exactly these four countries. therefore in my opinion the authors should limit describe the situation in the four countries.  Alternatively, it would be relevant to elaborate on what exactly comparing the different countries tells us.

In section 1.4 we have tried to be more explicit about the rationale behind the four countries we selected for this study.

As we stated, “the selection of these four countries had practical and theoretical reasons: (1) as part on an International Training Network (MultiMmind) these four countries were easily accessible through network members, (2) the four countries represent two different languages and, both between and within languages (three of the four countries are German-speaking) a variety of different contexts with respect to migration historical and current situation, organizational and educational regulations, health service delivery policies. This allows us to disentangle the effects of the different socio-organizational variables from possible effects of linguistic differences.”  We hope our point is now clearer.

  1. Comment by Reviewer 2

results.  For the sake of clarity the authors might give subtitles to the various paragraphs so that the reader can follow each objective and the corresponding result more easily.

in general, the results would need more structure.

We have now added further subtitles to the paragraphs in the result sections and hope it makes it easier to link each result with the respective framework.

  1. Comment by Reviewer 2

my biggest concern about the data analysis is that the study was conducted in the 4 countries but then all the data were put together. what justifies this.? are the performances similar? Then there are analyses comparing the 4 countries. What does this indicate (see also my point about the method)?

We see the Reviewer's point; however, the focus of our analyses was not on the comparisons of countries, but rather on the   systemic (organizational and social) factors that may influence SLTs’ attitudes and opinions towards multilingualism. The comparison of the different countries was just one of the means to disentangle such factors but comparing SLTs’ procedures and attitudes across countries was beyond the scope of our study. For this reason, we would prefer to keep the structure of the Results section as it is. We have tried to make this goal clearer in section 1.4.

  1. Comment by Reviewer 2

Finally, I would like to invite the authors to consider the possibility of conducting analyses correlating the answers to the questionnaire and the personal characteristics of the participants, such as age, gender, type of work (public/private), country of origin. in my opinion, this type of analysis could further enrich the very interesting results of this study and could favour some interpretations that the authors have already provided.

We fully agree with the Reviewer on this point. Unfortunately, for the sake of complete anonymity of our participants we had not collected all these demographic data. We have tried to make this clearer in the limitation section (4.5) of the paper. Nonetheless, we have analyzed responses considering country of workplace as one of the relevant characteristics of the respondents (along with experience, caseload, educational pathways etc.).

In fact, the type of service (public/private) was assessed, but in Italy only. Since the different types of health service organization in Italy do not reflect any similar typology outside the country, and since the discussion of the differences between such systems would imply addressing very specific organizational models, we felt this would bring us too far from the scope of our study. Moreover, since we found that service-related differences were strongly linked to differences in experience with multilingual children, and since different types of systems are unevenly spread across the country, we thought that interpretation of such results would become too complex. Nonetheless, we agree with the Reviewer that it would be interesting to analyze such differences more in-depth, and we plan to do so for a national publication focused only on the Italian system.

  1. Comment by Reviewer 2

The discussion should be revised on the basis of the changes made in the other sections.

Since the general structure of the manuscript was maintained, the discussion was mostly not changed. Nonetheless, we added some parts in the limitation section addressing the lack of more detailed personal information about respondents, (4.5) and in the practical implications section (4.6).

We hope and believe to have addressed all points and cleared all ambiguities.

We thank you for your insightful suggestions, which have – we think – been helpful for improving the manuscript.

Thank you for your attention,

Theresa Bloder, on behalf of all authors
